# Holistic Transfer: Towards Non-Disruptive Fine-Tuning with Partial Target Data

**Cheng-Hao Tu**[*1], **Hong-You Chen**[*1], **Zheda Mai**[1], **Jike Zhong**[1], **Vardaan Pahuja**[1],

**Tanya Berger-Wolf**[1], **Song Gao**[2], **Charles Stewart**[3], **Yu Su**[1], **Wei-Lun Chao**[1]

[1]The Ohio State University, [2]University of Wisconsin-Madison, [3]Rensselaer Polytechnic Institute

## Abstract

We propose a learning problem involving adapting a pre-trained source model to the target domain for classifying all classes that appeared in the source data, using target data that covers only a partial label space. This problem is practical, as it is unrealistic for the target end-users to collect data for all classes prior to adaptation. However, it has received limited attention in the literature. To shed light on this issue, we construct benchmark datasets and conduct extensive experiments to uncover the inherent challenges. We found a dilemma — on the one hand, adapting to the new target domain is important to claim better performance; on the other hand, we observe that preserving the classification accuracy of classes missing in the target adaptation data is highly challenging, let alone improving them. To tackle this, we identify two key directions: 1) disentangling domain gradients from classification gradients, and 2) preserving class relationships. We present several effective solutions that maintain the accuracy of the missing classes and enhance the overall performance, establishing solid baselines for holistic transfer of pre-trained models with partial target data.

## 1 Introduction

We are entering an era in which we can easily access pre-trained machine-learning models capable of classifying many objects. In practice, when end-users deploy these models in diverse domains, adaptation is often necessary to achieve high accuracy. To address this issue, considerable efforts have been devoted to domain adaptation (DA), aiming to transfer domain knowledge from the source to the target. This involves updating the model on an adaptation set collected from the target environment. Although various DA settings have been proposed to tackle different scenarios [75, 56, 55], we argue that previous setups rely on a strong assumption that hinders their applicability in the real world: the adaptation set contains samples from all possible labels (e.g., classes) in the target environments. Taking classification as an example, to adapt a pre-trained source model's recognition ability over $C$ classes to a novel domain, DA necessitates access to a target dataset encompassing all $C$ classes.

In reality, the adaptation set may only have samples of much fewer classes $C' < C$ due to limited sample size, collections bias, etc. Many pertinent examples can be found in realistic AI applications. Considering wildlife monitoring [38] as an instance, where data collection often occurs passively, through smart camera traps, awaiting animal appearances. Consequently, when a smart camera trap is deployed to a new location and requires adaptation, assembling a comprehensive target dataset containing all the animal species becomes a daunting task, especially if a specific rare species does not appear within the data collection period due to seasonal variations or animal migrations.

---

[*]Equal contributions

Table 1: **Motivating examples of Holistic Transfer (HT).** We propose Holistic Transfer, a novel and realistic transfer learning problem. Unlike traditional paradigms, HT focuses on scenarios where only a subset of classes from the target test environment is present in the target training set. Naive fine-tuning (FT) here can be disruptive.

| Dataset | Scenario | Source | Target Training | Target Test |
|---|---|---|---|---|
| Office-Home [74] | Domain shift | Real (65 classes) | Clipart (30 classes) | Clipart (65 classes) |
| FEMNIST [4] | Personalization on scarce data | Many writers: 0-9, a-z | Writer X: a, 5 | Writer X: b, 5, e, x |
| iWildCam [38] | Camera traps in the wild | Many locations | Location N: before 2010 | Location N: after 2010 |
| VTAB [91] | FT zero-shot models | CLIP's training data | Caltech101 (50 classes) | Caltech101 (101 classes) |
| iNaturalist (Fungi) [73] | Confusion classes | CLIP's training data | Non-toxic | Non-toxic, toxic |

To underscore the significance of the problem we address in this paper, it is worth noting that the literature has made significant progress towards training a general recognition model that can predict almost any classes in the vocabulary space such as CLIP [60] and ALIGN [35]. However, we argue that the current DA approaches applied to these pre-trained models face a practical learning dilemma in adapting the versatile classification capability. On the one hand, adaptation to the target domain style is preferred. On the other hand, the models adapted on *target training data* whose labels cover a subset of the *target test data* can lead to risky outcomes for the end users. Through experiments, we observed that fine-tuning the model with empirical risk minimization on such partial target data, which fails to represent the overall target distribution, hampers the performance of classes that were *already pre-trained but unavailable during target adaptation*.

This is frustrating for both parties as a lose-lose dilemma: on the pre-training side, it negates the extensive effort invested in preparing for such a source model as the foundation for adaptation. On the side of downstream users, despite the collection of a target dataset for adaptation purposes, the difficulty of covering all target classes results in an adapted model performing *even worse* than the source model in the actual target environment. In response to the dilemma, we ask: *can we transfer more holistically, i.e., improving the overall performance without hurting missing class performance?* We propose a novel and realistic transfer learning paradigm that aims to transfer the source model's discriminative capabilities to the target domain — specifically, the task of classifying all $C$ objects in the source domain — while working with a target training dataset comprising only a limited subset of these classes, i.e., $C' < C$. We refer to the problem as **Holistic Transfer (HT)**.

**Contributions.** We formulate a new learning paradigm called Holistic Transfer (HT).

1. We define a new, practical transfer learning paradigm that complements existing ones (section 2).
2. We construct extensive benchmarks in subsection 2.2 and section 4 to support HT studies.
3. We systematically explore methods to approach HT in section 3 and provide strong baselines.
4. Our extensive experiments shed insights towards solving HT (section 4 and section 5).

## 2 Holistic Transfer Learning

### 2.1 Problem Definition

We propose a novel and realistic transfer learning paradigm called **Holistic Transfer (HT)**.

**Setup.** We focus on the task of multi-class classification. Let us consider a source model parametrized by $\boldsymbol{\theta}_{\mathcal{S}}$. It is pre-trained on a source dataset[1] $\mathcal{S} = \{(\boldsymbol{x}_i^{\mathcal{S}}, y_i^{\mathcal{S}})\}_{i=1}^{|\mathcal{S}|}$, where $\boldsymbol{x}_i^{\mathcal{S}}$ is the input (e.g., an image) and $y_i^{\mathcal{S}}$ is the corresponding label (e.g., a class of the $C$ classes in the label space).

---

[1]As will be introduced in subsection 2.2, the concept of the "source" domain can be quite general: a dataset with different input style, data from a collection of different users, a general pre-training dataset, etc.

Let the joint distribution of the source be $P_S$ and the true distribution of the target environment be $P_{\mathcal{T}^\star}$. The goal of HT, just like standard domain adaptation, is to leverage the source model to learn a target model $\boldsymbol{\theta}_\mathcal{T}$ that performs well on the true target data, measured by a loss $\mathcal{L}$.

**Remarks: Holistic Transfer.** Typical transfer learning setups assume the target training data $\mathcal{T}$ are sampled IID from $P_{\mathcal{T}^\star}$. However, this may not hold in practical scenarios due to sampling bias or insufficient sample size, making the target training data $\mathcal{T}$ biased; i.e., $P_\mathcal{T} \neq P_{\mathcal{T}^\star}$. The target training data $\mathcal{T} = \{(\boldsymbol{x}_i^\mathcal{T}, y_i^\mathcal{T})\}_{i=1}^{|\mathcal{T}|}$ from the target domain are limited to a subset of classes $C' < C$.

At first glance, our setting seems quite similar to partial DA [5], which also considers the situation of partial target classes (i.e., $C' < C$). However, its goal is to perform well only on those $C'$ classes after adaptation. In contrast, our setting aims to adapt the source model's holistic capability of recognizing all $C$ classes to the target domain. Below we first give the HT definition grounding on the literature formally. We will contrast the HT problem with other existing learning paradigms in subsection 2.3.

**Assumptions of $P_S$ vs. $P_{\mathcal{T}^\star}$.** Following standard domain adaptation [75, 56, 55], we consider $P_S$ and $P_{\mathcal{T}^\star}$ have covariate shifts; thus the target features need to be adapted from the source ones to tailor the target domain style, i.e., $P_S(\boldsymbol{x}|y) \neq P_{\mathcal{T}^\star}(\boldsymbol{x}|y)$. Typically, the label space of the true target domain is already covered by the source counterpart, or $\mathcal{Y}_{\mathcal{T}^\star} \subseteq \mathcal{Y}_S$.

**Assumptions of $P_\mathcal{T}$ vs. $P_{\mathcal{T}^\star}$.** We study the setting that $\mathcal{T}$ is a (potentially biased) sampled dataset from $P_{\mathcal{T}^\star}$, therefore, $P_\mathcal{T}$ and $P_{\mathcal{T}^\star}$ should share the same domain styles and semantics. That is, $P_\mathcal{T}(\boldsymbol{x}|y) \approx P_{\mathcal{T}^\star}(\boldsymbol{x}|y), P_\mathcal{T}(y|\boldsymbol{x}) \approx P_{\mathcal{T}^\star}(y|\boldsymbol{x})$. However, the label space of $P_\mathcal{T}$ might be incomplete w.r.t. $P_{\mathcal{T}^\star}$; $P_\mathcal{T}(y) \neq P_{\mathcal{T}^\star}(y)$. In other words, the test set $\mathcal{T}^\star$ sampled from $P_{\mathcal{T}^\star}$ might contain data belonging to classes seen in $S$ but not appear in the target training set $\mathcal{T}$. The ultimate goal of the target model is to perform well on $P_{\mathcal{T}^\star}$.

**Objective.** The objective of HT is to learn a predictor $f(\cdot; \boldsymbol{\theta}_\mathcal{T}) : \mathcal{X}_{\mathcal{T}^\star} \to \mathcal{Y}_{\mathcal{T}^\star}$ that minimizes the expected loss on the true target distribution $P_{\mathcal{T}^\star}$ but *given only the partial observations of a target training dataset $\mathcal{T}$ sampled from $P_\mathcal{T}$ without accessing to $P_{\mathcal{T}^\star}$*:

$$\mathcal{R}(f; P_{\mathcal{T}^\star}) = \mathbb{E}_{(\boldsymbol{x}, y) \sim P_{\mathcal{T}^\star}} [\mathcal{L}(y, f(\boldsymbol{x}))]; \qquad \hat{\mathcal{R}}(f; \mathcal{T}) = \sum\nolimits_{\{(\boldsymbol{x}_i, y_i)\}_{i=1}^{|\mathcal{T}|}} \mathcal{L}(y_i, f(\boldsymbol{x}_i)). \quad (1)$$

That is, we can only compute the empirical risk $\hat{\mathcal{R}}(f; \mathcal{T})$ on $\mathcal{T}$ but not the true risk $\mathcal{R}(f; P_{\mathcal{T}^\star})$.

**Challenges in a realistic HT scenario.** We focus on a more realistic scenario of HT that comes with the following practical constraints and properties when solving $\mathcal{R}(f; P_{\mathcal{T}^\star})$ in Equation 1.

- **Missing classes in target training data.** The main challenge is that some classes might only appear in $P_{\mathcal{T}^\star}$ in testing but not yet seen in the target training set $\mathcal{T}$. Minimizing $\hat{\mathcal{R}}(f; \mathcal{T})$ naively might be risky since it is biased estimation of $\mathcal{R}(f; P_{\mathcal{T}^\star})$.
- **Disparate impact [26].** Training on the partial target data $\mathcal{T}$ can result in disproportionate effects on the performance of unseen target classes in testing. The impact may vary based on the semantic affinity between classes, potentially causing serious false negatives. For instance, if a biologist fine-tunes an ImageNet classifier on some non-toxic plants and uses it to recognize plants in the wild, it will likely misclassify those visually similar toxic plants as non-toxic ones.
- **Covariate shifts from the source domain.** If we only have $\mathcal{T}$, the missing classes are very difficult to be predicted as it is zero-shot classification. A more practical scenario is to leverage an external source classifier that has been pre-trained on $S$ with many classes. This relaxes the problem to be a special (supervised) domain adaptation task that leverages the source classifier's discriminative ability (as $P_S(y|\boldsymbol{x}) \approx P_{\mathcal{T}^\star}(y|\boldsymbol{x})$) while adapting the input styles $P_S(\boldsymbol{x}) \neq P_{\mathcal{T}^\star}(\boldsymbol{x})$.
- **Source-data-free transfer.** Last but not least, the source model $\boldsymbol{\theta}_S$ is available but the source dataset $S$ is not. Due to privacy concerns or the dataset's large size, accessing the source dataset is often unrealistic, especially for foundational models pre-trained in large scales like CLIP [60].

Therefore, we believe HT is an important and challenging problem to approach for improving towards safer transfer learning in the wild, *given only a source classifier $\boldsymbol{\theta}_S$ and a partial target dataset $\mathcal{T}$*.

## 2.2 Benchmarks Curation for Holistic Transfer: Motivating Examples

To support the study of the HT problem, we create a benchmark that covers extensive scenarios across both experimental and realistic public datasets. More details about dataset curation are in the supplementary. As introduced in Table 1, HT is applicable in various use cases, such as:

- Office-Home [74]: domain adaptation while some classes are missing in the target training set.

- FEMNIST [4]: personalized hand-written alphanumeric recognition with writers' styles. Training samples of a single writer are insufficient to cover all classes.
- iWildCam [38]: species recognition across camera traps of many geo-locations. Adaptation is based on images collected so far, and is deployed for the future; i.e., samples are biased temporally.
- VTAB [91]: fine-tuning zero-shot models for diverse vision tasks with partial classes each.
- iNaturalist (2021 version, Fungi) [73]: classification of visually-similar poisonous fungi.

Each task pre-trains a source model on the source domain, transfers to the target domain with only partial classes, and tests on the full-class target test data. For the Office-Home dataset, the source model is trained on one domain different from the target domain. For the FEMNIST/iWildCam datasets, the source and target data are split by writers/locations. For the VTAB and iNaturalist datasets, there are no explicit domains. A general dataset can be used for pre-training the source model. Here we will use CLIP [60] just as an example.

## 2.3 Comparison to Other Paradigms: To Adapt or Not to Adapt, and How?

The most common way of transfer learning is probably fine-tuning, i.e., initializing the target model with the source model parameters and optimizing it with empirical risk minimization (ERM) on the target data. In the HT setup, fine-tuning by minimizing $\hat{\mathcal{R}}(f; \mathcal{T})$ in Equation 1, on the one hand, improves on the performance of those classes seen in the target training set $\mathcal{T}$. On the other hand, it can be disruptive in terms of the true risk $\mathcal{R}(f; P_{\mathcal{T}^\star})$ due to catastrophic forgetting [37] of those classes *already seen in the source domain* but *unseen during the target fine-tuning*. This creates a dilemma for the user about whether to adapt or not. As shown in the example in Figure 1 (details in section 4), naive fine-tuning does improve those seen classes over the source model while unseen classes can degrade drastically. On the contrary, our proposed treatment (see section 3) can maintain the unseen performance and improve the seen ones, compared to the source model (the 0 epoch).

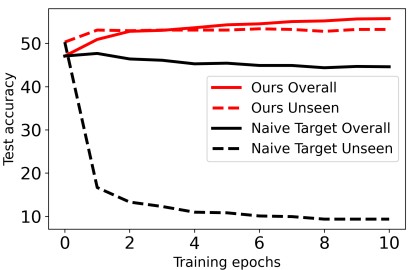

Figure 1: **Fine-tuning disrupts unseen target class performance.** An HT example of Ar → Cl domains on Office-Home dataset.

We note the HT problem we proposed is a unique machine learning paradigm that lies between and *complementary* to existing learning-with-distribution-shift problems [59]. From a high-level and simplified view, many of them involve three stages: 1) source training on $\mathcal{S}$ to obtain a source model, 2) target training on $\mathcal{T}$, then 3) evaluation of the target goal distribution $P_{\mathcal{T}}^{test}$.

We summarize the difference to some popular paradigms in Table 2, including general transfer learning and (supervised) domain adaptation (DA) [98, 75] that assume the target distributions in training and testing are matched. Continual learning (CL) [37, 51, 47] that typically disallows access to source data during target training, aiming to build a multi-tasking model that performs well on all the (possibly many) distributions that have been trained on so far. They all assume the training and test data are matched in distributions ($P_{\mathcal{T}} \approx P_{\mathcal{T}}^{test}$). Some recent works about OOD domain generalization [41, 3, 79] study fine-tuning from a versatile pre-trained model and preserving its robustness in testing to different unseen input styles of the seen classes but not for unseen classes.

As illustrated in Figure 2 (d), HT requires a very different ability (the green arrow): "generalize" the style shifts learned on the target seen classes to the classes unseen during target training (the red dashed arrow). Compared to OOD generalization, our goal is much harder since it requires transferring the source model to the target styles, for classes both seen and unseen in the target training data. Given only seen classes, using fine-tuning here is notoriously biased to them and disrupts the unseen performance. HT is also addressing quite different challenges from source-free DA [48], unsupervised DA (with partial classes [32]), and test-time adaptation [69] which focus on recovering the true labels for the unlabeled target data. More related works are in the supplementary.

**Our goals.** To this end, we raise the following research questions:

1. Can we resolve the adaptation dilemma and transfer holistically from the source model, improving the test performance of classes seen in the target training set while not hurting the unseen ones?
2. Is it possible to generalize the style shifts to unseen classes? Or how far are we?
3. When should HT work and not work in realistic cases?

Table 2: **Comparison of different paradigms: Source Training→Target Training→Test on Target Goal.** Let $P_\mathcal{S}$ and $P_\mathcal{T}$ be the distributions of the source data $\mathcal{S}$ and the target training set $\mathcal{T}$, respectively. $P_\mathcal{T}^{test}$ is the target test distribution as the final goal. $P_{\mathcal{T}^\star}$ is the distribution that may have certain discrepancy from $P_\mathcal{T}$.

| Settings | Target Training | Target Goal $P_\mathcal{T}^{test}$ | $P_\mathcal{S}$ vs. $P_\mathcal{T}^{test}$ | $P_\mathcal{T}$ vs. $P_\mathcal{T}^{test}$ |
|---|---|---|---|---|
| Standard transfer | $\boldsymbol{\theta}_\mathcal{S}$ and/or $\mathcal{S}, \mathcal{T}$ | $P_\mathcal{T}$ | $P_\mathcal{S} \neq P_\mathcal{T}^{test}$ | $P_\mathcal{T} \approx P_\mathcal{T}^{test}$ |
| Domain adaptation | $\boldsymbol{\theta}_\mathcal{S}$ and/or $\mathcal{S}, \mathcal{T}$ | $P_\mathcal{T}$ | $P_\mathcal{S}(y\|\boldsymbol{x}) \approx P_\mathcal{T}^{test}(y\|\boldsymbol{x})$ | $P_\mathcal{T} \approx P_\mathcal{T}^{test}$ |
| Continual learning | $\boldsymbol{\theta}_\mathcal{S}, \mathcal{T}$ | $P_\mathcal{S} + P_\mathcal{T}$ | $P_\mathcal{S} \neq P_\mathcal{T}^{test}$ | $P_\mathcal{T}^{test} \approx P_\mathcal{S} + P_\mathcal{T}$ |
| OOD generalization | $\boldsymbol{\theta}_\mathcal{S}, \mathcal{T}$ | $P_{\mathcal{T}^\star}$ | $P_\mathcal{S} \neq P_\mathcal{T}^{test}$ | $P_\mathcal{T}(\boldsymbol{x}\|y) \neq P_{\mathcal{T}^\star}(\boldsymbol{x}\|y),$ $P_\mathcal{T}(y) \approx P_{\mathcal{T}^\star}(y)$ |
| Holistic transfer | $\boldsymbol{\theta}_\mathcal{S}, \mathcal{T}$ | $P_{\mathcal{T}^\star}$ | $P_\mathcal{S}(y\|\boldsymbol{x}) \approx P_\mathcal{T}^{test}(y\|\boldsymbol{x})$ | $P_\mathcal{T}(\boldsymbol{x}\|y) \approx P_{\mathcal{T}^\star}(\boldsymbol{x}\|y),$ $P_\mathcal{T}(y) \neq P_{\mathcal{T}^\star}(y)$ |

(a) Domain Adaptation    (b) Continual Learning    (c) OOD Generalization    (d) Holistic Transfer

Figure 2: **Different settings of learning with distribution shift.** Consider a data distribution space of different styles (**x-axis**) and classes (**y-axis**), each algorithm leverages the source data (the blue region) to facilitate the knowledge transfer (the green arrow) to the target training data (the yellow region). The target test distribution $P_\mathcal{T}^{test}$ is highlighted with the **black box**. The red arrow represents the generalization for unseen test data.

## 3 Towards Non-Disruptive Fine-Tuning

### 3.1 Solving the HT Objective

We now discuss approaching the HT objective of Equation 1. The key difficulty is the fact that the true risk $\mathcal{R}(f; P_{\mathcal{T}^\star})$ and the empirical risk $\hat{\mathcal{R}}(f; \mathcal{T})$ are in different distributions (i.e., $P_{\mathcal{T}^\star}(y) \neq P_\mathcal{T}(y)$), such that methods relying on ERM fine-tuning can be seriously biased to seen classes, as discussed in subsection 2.3. Let $P_\mathcal{T}$ has $C$ labels $\mathcal{Y}_\mathcal{T} = \{y_1^{\text{Seen}}, ..., y_C^{\text{Seen}}\}$ and $P_{\mathcal{T}^\star}$ has some extra unseen labels $\mathcal{Y}_{\mathcal{T}^\star} = \mathcal{Y}_\mathcal{T} \cup \{y_{C+1}^{\text{Unseen}}, y_{C+2}^{\text{Unseen}}, ..., y_{C^\star}^{\text{Unseen}}\}$. Worth noting, we fairly assume the source data cover all target classes $\mathcal{Y}_{\mathcal{T}^\star} \subseteq \mathcal{Y}_\mathcal{S}$; if a test class was not seen in $\mathcal{S}$, it becomes a zero-shot problem which might not be solved by transfer learning.

**Oracle solutions.** An oracle solution is to directly match the distributions, making $P_{\mathcal{T}^\star}(y) = P_\mathcal{T}(y)$ by augmenting data to $\mathcal{T}$. For instance, one can collect more data for those unseen classes in the target domain, or translate the source data sample of an unseen class $(\boldsymbol{x}^\mathcal{S}, y_C^{\text{Unseen}})$ into $(\boldsymbol{x}^\mathcal{T}, y_C^{\text{Unseen}})$ with a style transfer model from $\mathcal{S}$ to $\mathcal{T}$. They are obviously not available for HT— the former requires the cost to collect complete labeled data in the target (essentially the upper bound in domain adaptation); the latter needs to access the source data which is unrealistic as discussed in subsection 2.1.

**Suboptimal solutions.** By leveraging techniques in other paradigms we discussed in subsection 2.3, it might be possible to improve over plain fine-tuning. However, none explicitly recovers the joint probability of the target goal distribution $P_{\mathcal{T}^\star}(\boldsymbol{x}, y)$, but only $P_\mathcal{T}(\boldsymbol{x}, y)$ or $P_\mathcal{T}(y)$.

Next, we propose two directions that together could approach $P_{\mathcal{T}^\star}(\boldsymbol{x}, y)$ under HT constraints.

### 3.2 Disentangling Covariate Shifts from Disruptive Concept Shifts

Minimizing $\hat{\mathcal{R}}(f; \mathcal{T})$ on $\mathcal{T}$ transfers the model from $P_\mathcal{S}(\boldsymbol{x}, y; \boldsymbol{\theta}_\mathcal{S})$ to $\hat{P}_\mathcal{T}(\boldsymbol{x}, y; \boldsymbol{\theta}_\mathcal{T})$ which optimizes towards the target styles $P_\mathcal{T}(\boldsymbol{x}|y)$ (which should be the same as $P_{\mathcal{T}^\star}(\boldsymbol{x}|y)$) but also collapses the label distribution to $\hat{P}_\mathcal{T}(y; \boldsymbol{\theta}_\mathcal{T})$. The model will unlikely predict a $y_C^{\text{Unseen}}$ for any inputs. If we can only adapt the covariates from the source classifier, then we have

$$\hat{P}_{\mathcal{T}^\star}(\boldsymbol{x}, y; \boldsymbol{\theta}_\mathcal{T}) := \hat{P}_{\mathcal{T}^\star}(y|\boldsymbol{x}; \boldsymbol{\theta}_\mathcal{T})\hat{P}_{\mathcal{T}^\star}(\boldsymbol{x}; \boldsymbol{\theta}_\mathcal{T}), \tag{2}$$

$$\leftarrow \hat{P}_{\mathcal{T}^\star}(y|\boldsymbol{x}; \boldsymbol{\theta}_\mathcal{S})P_\mathcal{S}(\boldsymbol{x}; \boldsymbol{\theta}_\mathcal{S}), \quad [\text{Style Adaptation } \hat{P}_{\mathcal{T}^\star}(\boldsymbol{x}; \boldsymbol{\theta}_\mathcal{T}) \leftarrow P_\mathcal{S}(\boldsymbol{x}; \boldsymbol{\theta}_\mathcal{S})] \tag{3}$$

$$\leftarrow P_\mathcal{S}(y|\boldsymbol{x}; \boldsymbol{\theta}_\mathcal{S})P_\mathcal{S}(\boldsymbol{x}; \boldsymbol{\theta}_\mathcal{S}). \quad [P_{\mathcal{T}^\star}(y|\boldsymbol{x}) \approx P_\mathcal{T}(y|\boldsymbol{x}) \approx P_\mathcal{S}(y|\boldsymbol{x})] \tag{4}$$

Equation 3 will hold if $P_{\mathcal{T}^\star}(\boldsymbol{x}|y) \approx P_{\mathcal{T}}(\boldsymbol{x}|y)$. However, how can we disentangle the covariate shifts from $\nabla\hat{\mathcal{R}}(f;\mathcal{T})$ and change the style to $P_{\mathcal{T}}(\boldsymbol{x})$ only without concept shift [40]? We explore:

- **Updating batchnorm statistics only.** The statistics in batchnorm layers (in feed-forward networks) are believed to capture the domain information. Updating them is often considered as a strong baseline in DA [46]. It infers on $\mathcal{T}$ and re-calculates the feature means and variances, which is less likely to be impacted by the missing classes issue.
- **Training extra instance normalization layers.** We explore another way based on instance normalization (IN) [72], which normalizes within each input itself over its spatial dimension. IN is widely used in style transfer to capture the input styles without changing their semantics. We examine a way for HT by inserting extra IN layers into the source model (which may not have IN layers originally) and fine-tuning them with other DNN parameters fixed to prevent concept shifts.

Unfortunately, we found updating normalization layers is not yet satisfying — receiving limited improvements and still suffering from catastrophic forgetting. We thus hypothesize the key is to consider the optimization besides architectures.

**Proposed Leave-Out Local SGD (LOLSGD).** In our HT problem, fine-tuning is affected by the covariate shift (from the source to the target) and the disruptive concept shift (from classifying all the classes to classifying the seen classes). We aim to disentangle them or, more precisely, reduce the disruptive concept shift by subsampling multiple datasets that each contain a subset of the seen classes.

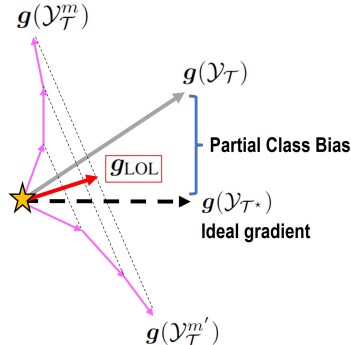

Figure 3: **LOLSGD illustration.**

We propose a novel approach based on local SGD [68] in distributed learning. The local SGD gradient is accumulated over several consecutive SGD steps, possibly many runs in parallel. For notations, let the local SGD gradient computed on the samples that their labels are inside a label space $\mathcal{Y}$ be $\boldsymbol{g}(\mathcal{Y})$. If sampling IID from $\mathcal{T}$ and the local step is 1, it reduces to standard SGD. Instead of using $\boldsymbol{g}(\mathcal{Y}_{\mathcal{T}})$, which will bias to the seen target classes $\mathcal{Y}_{\mathcal{T}}$, our idea is to mitigate the change towards the label and encourage the change in styles.

Inspired by recent theoretical analyses in meta-representation learning [21, 20, 12, 13], we intentionally sample $m$ non-IID subsets of labels $\mathcal{Y}_{\mathcal{T}}^m \subsetneq \mathcal{Y}_{\mathcal{T}}$ by *leaving out* some classes, compute $\boldsymbol{g}(\mathcal{Y}_{\mathcal{T}}^m)$ for each $m$, and average their local gradients $\boldsymbol{g}(\mathcal{Y}_{\mathcal{T}}^m)$

$$\boldsymbol{g}_{\text{LOL}}(\mathcal{Y}_{\mathcal{T}}; M) := \frac{1}{M}\sum_m \boldsymbol{g}(\mathcal{Y}_{\mathcal{T}}^m), \; \boldsymbol{g}(\mathcal{Y}_{\mathcal{T}}^m) := \boldsymbol{\theta} - \arg\min_{\boldsymbol{\theta}} \sum\nolimits_{\{i \in |\mathcal{T}| \; | \; y_i \in \mathcal{Y}_{\mathcal{T}}^m\}} \mathcal{L}(y_i, f(\boldsymbol{x}_i; \boldsymbol{\theta})). \quad (5)$$

During training, each update (with size $\eta$) becomes $-\eta\boldsymbol{g}_{\text{LOL}}$ instead of $-\eta\nabla\hat{\mathcal{R}}(f;\mathcal{T})$. Intuitively, $\boldsymbol{g}(\mathcal{Y}_{\mathcal{T}}^m)$ is biased towards the respective classes $\mathcal{Y}_{\mathcal{T}}^m$ and drifts away from each other along more local steps, similar to the model shift problem in decentralized learning [29, 39, 10]. As a blessing here (rather than a curse!), our key insight is averaging them could "cancel out" the gradients biased to certain classes. The style should be adapted for Equation 3 without conflicts since all local gradients are directed towards the same target style, as illustrated in Figure 3. When updating the model separately in parallel with these subsampled datasets, each updated model is affected by a different concept shift but a shared covariate shift. Averaging these models can potentially cancel out the disruptive concept shifts and strengthen the shared covariate shift. We verify our intuition in Table 19 in the supplementary. Compared to $\nabla\hat{\mathcal{R}}(f;\mathcal{T})$, $\boldsymbol{g}_{\text{LOL}}$ averages $M$ ($M = 10$ in our experiments) accumulated local gradients thus more expensive to compute. As will be shown by experiments, LOLSGD can outperform standard SGD with similar computation budgets in fair comparisons.

LOLSGD is fundamentally different from meta-learning, whose goal is to learn a meta-modal that can be easily applied to a future task (e.g., a few-shot classification task). While meta-learning also subsamples its meta-training set, it is mainly to simulate multiple future tasks for learning the meta-model, not to cancel out unwanted gradients. LOLSGD is also an extension of local SGD. The goal of local SGD is mainly to reduce the communication overhead of large-scale distributed training. In contrast, in LOLSGD, the target training data set is not decentralized initially, but we strategically subsample it to simulate different concept shifts.

## 3.3 Preserving Class Relationships

The subsection 3.2 assumes we can change the styles while not changing the conditional probability of the source model $P_{\mathcal{S}}(y|\boldsymbol{x})$ in Equation 4. We observe in practice it can still be sensitive and lean towards $\{y^{\text{Seen}}\}$. We, therefore, hypothesize that it is necessary to have some regularization to preserve the nice class relationships encoded in the source model already during target training such that Equation 4 could hold. Let us decompose the predictor $f(\boldsymbol{x};\boldsymbol{\theta}) = g(\boldsymbol{z};\boldsymbol{w}), \boldsymbol{z} = h(\boldsymbol{x};\boldsymbol{\phi})$ to be a feature extractor $h$ and a linear classifier $g$. We explore the following strategies:

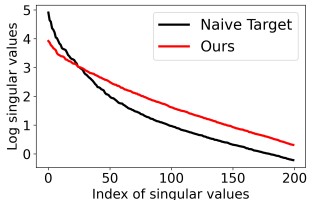

Figure 4: **Feature rank collapse.** Top singular values of Office-Home adapted features.

- **Frozen linear classifier.** Even the source features is already perfect for $P_{\mathcal{S}}(y|\boldsymbol{x}) = P_{\mathcal{T}^\star}(y|\boldsymbol{x})$, it can collapse quickly as the logits of $y^{\text{Unseen}}$ will be suppressed by $\nabla\hat{\mathcal{R}}(f;\mathcal{T})$. We found it crucial to freeze the linear classifier $\boldsymbol{w}$, which corresponds to the common practice in source-free DA [48].
- **Selective distillation.** Another way is to utilize distillation that can prevent forgetting in CL [47, 93, 22], forcing the soft labels of the target model and the source model to match. Given a sample $(\boldsymbol{x}, y^{\text{Seen}}) \in \mathcal{T}$ and let $f^{\text{Unseen}}(\boldsymbol{x})$ be the logits of $y^{\text{Unseen}}$ dimensions. Here, only the unseen class logits need to be distilled since seen class logits are reliable as the target model is trained on ground truths, unlike source seen class logits (which are suboptimal due to domain shift). We add a distillation loss $\mathcal{L}_{\text{distill}} = \text{KL}(\sigma(f^{\text{Unseen}}(\boldsymbol{x};\boldsymbol{\theta}_{\mathcal{S}}))||\sigma(f^{\text{Unseen}}(\boldsymbol{x};\boldsymbol{\theta}_{\mathcal{T}})))$ with the Kullback–Leibler (KL) divergence, where $\sigma(\cdot)$ is the softmax function.
- **Feature rank regularization.** A key observation is in the feature dynamic of fine-tuning on the partial target data — instead of shifting to another full-rank space, we found the features tend to collapse to a low-dimensional space, as shown in Figure 4 — implying the target model "shortcutting" the solution and hampering the generalization ability [76] to handle unseen target classes. We consider a regularizer to avoid too many singular values of the features becoming zeros. Given the feature matrix of a mini-batch $\boldsymbol{Z} \in \mathbb{R}^{N \times d}$ and its mean feature $\bar{\boldsymbol{z}} \in \mathbb{R}^d$, we compute its covariance matrix $\boldsymbol{C} = \frac{1}{N}\sum_{n=1}^{N}(\boldsymbol{z}_n - \bar{\boldsymbol{z}})(\boldsymbol{z}_n - \bar{\boldsymbol{z}})^{\mathsf{T}}$ and a regularizer $\mathcal{L}_{\text{rank}} = \|\text{diag}(\boldsymbol{C}^{\mathsf{T}}\boldsymbol{C})\|_2^2$, inspired by [36, 17, 66].

We note that, these techniques only preserve the source class relationships, but they do not improve the transfer to the target. Thus they should be considered together with ERM or LOLSGD for HT.

**Ensemble with the source model.** As pointed out by [79], a way to reclaim some ability of the source model is to average the parameters of the source and target models, assuming they lie in a linearly-connected space. We consider it complementary post-processing for the target training techniques we considered above. We relax the assumption and further examine the ensemble of the source and target models in their predictions (after the softmax function $\sigma$ for logits).

## 4 Experiments

**General setup.** To answer the questions we raised in subsection 2.3, we experiment on extensive HT scenarios proposed in subsection 2.2. For each task, we pre-train a source model in standard practice, discard the source data, and adapt the source parameters $\boldsymbol{\theta}_{\mathcal{S}}$ to obtain the target model. We split the target data for training and testing, based on different scenarios. The training and test sets share the same styles but cover different sets of classes, where some classes are unseen in training. We assume the source model covers all classes and the goal is to adapt to the target domain for both the seen and unseen classes. All methods use the cross-entropy loss for $\mathcal{L}$ and SGD momentum optimizer. All experiments fine-tune the source model for 20 epochs (10 for FEMNIST) by default. For the regularizers in section 3, we attach them as $\mathcal{L} + \lambda_{\text{dsitill (or rank)}}\mathcal{L}_{\text{dsitill (or rank)}}$, where the weights $\lambda$ are quite stable thus we did not search for it exhaustedly for every method, but use the same ones per dataset. For the proposed LOLSGD, we set $M = 10$ and randomly drop 3 classes when sampling $\mathcal{Y}_{\mathcal{T}}^m$ in Equation 5. Each subgradient in LOLSGD is by

Table 4: Effects of Source Ensemble (Avg. as in Table 3).

| Methods / Acc. | Overall | Unseen |
|---|---|---|
| Source | 63.90 | 65.65 |
| Naive Target (w/o SE) | 53.50 | 22.36 |
| Naive Target (WISE) | 53.56 | 25.26 |
| Naive Target (SE) | 67.46 | 50.89 |
| SGD (WISE) ❄ | 69.75 | 60.42 |
| SGD (SE) ❄ | 71.01 | 59.19 |
| SWA ❄ | 64.99 | 45.10 |
| SWAD ❄ | 65.05 | 45.30 |
| SGD $+\mathcal{L}_{\text{distill}}$ ❄ | 69.78 | 62.94 |
| SGD $+\mathcal{L}_{\text{rank}}$ ❄ | **71.05** | 65.48 |
| LOLSGD ❄ | 70.70 | 63.14 |
| LOLSGD $+\mathcal{L}_{\text{distill}}$ ❄ | 68.71 | 64.56 |
| LOLSGD $+\mathcal{L}_{\text{rank}}$ ❄ | 69.94 | **66.29** |
| **Best** $\Delta$Acc$_{\text{-Naive Target}}$ | 17.55 | 43.12 |

Table 3: Domain adaptation 65-way test accuracy on Office-Home with 30 seen and 35 unseen classes. Variances of random seeds are provided in the supplementary due to space limit. Blue: HT methods suggested by us in section 3. Red: methods that significantly improve overall accuracy and successfully maintain unseen accuracy on the source model. ❄: the linear classifier is frozen during training.

| Domains: source→target Methods / Acc. | Ar→Cl Overall | Unseen | Ar→Pr Overall | Unseen | Ar→Rw Overall | Unseen | Rw→Ar Overall | Unseen | Rw→Cl Overall | Unseen | Rw→Pr Overall | Unseen | Avg. Overall | Unseen |
|---|---|---|---|---|---|---|---|---|---|---|---|---|---|---|
| Source | 47.07 | 50.29 | 67.45 | 72.00 | 72.73 | 74.68 | 65.73 | 68.19 | 51.13 | 49.26 | 79.28 | 79.47 | 63.90 | 65.65 |
| Naive Target | 44.96 | 9.06 | 52.39 | 23.75 | 59.48 | 32.91 | 54.93 | 28.03 | 46.92 | 6.34 | 62.31 | 34.08 | 53.50 | 22.36 |
| BN only | 45.94 | 14.33 | 54.30 | 30.25 | 63.22 | 41.91 | 61.47 | 42.32 | 49.10 | 13.42 | 71.57 | 51.96 | 57.60 | 32.36 |
| BN (stats) only | 47.44 | 50.44 | 65.03 | 70.88 | 71.76 | 74.12 | 65.73 | 70.89 | 55.56 | 55.75 | 80.09 | 82.96 | 64.27 | 67.51 |
| IN only | 48.96 | 49.42 | 68.19 | 71.13 | 76.23 | 74.26 | 66.40 | 68.46 | 54.89 | 51.48 | 79.35 | 79.05 | 65.17 | 65.63 |
| LP-FT | 43.91 | 6.73 | 48.05 | 16.38 | 55.81 | 25.60 | 50.67 | 19.14 | 45.04 | 3.54 | 56.21 | 22.21 | 49.95 | 15.60 |
| SGD (w/ frozen classifier ❄) | 52.11 | 24.12 | 64.36 | 44.75 | 70.26 | 55.27 | 68.27 | 53.10 | 55.26 | 24.19 | 75.75 | 59.22 | 64.34 | 43.44 |
| SGD + $\mathcal{L}_{distill}$ ❄ | 56.54 | 39.18 | 72.81 | 61.63 | 75.73 | 67.51 | 70.13 | 64.15 | 61.96 | 51.45 | 80.60 | 70.53 | 69.63 | 57.41 |
| SGD + $\mathcal{L}_{rank}$ ❄ | 59.17 | 39.47 | 70.68 | 57.57 | 74.31 | 64.28 | 70.40 | 64.42 | 61.65 | 39.09 | 79.57 | 68.86 | 69.30 | 55.64 |
| SWA ❄ | 53.23 | 26.32 | 65.25 | 46.00 | 71.46 | 56.82 | 68.40 | 55.26 | 56.24 | 26.55 | 75.39 | 59.64 | 64.99 | 45.10 |
| SWAD ❄ | 53.38 | 26.46 | 65.25 | 46.00 | 71.61 | 57.24 | 68.53 | 55.53 | 55.87 | 26.40 | 75.68 | 60.20 | 65.05 | 45.30 |
| LOLSGD ❄ | 56.47 | 35.09 | 70.83 | 56.88 | 74.91 | 64.84 | 70.53 | 62.53 | 58.72 | 35.25 | 80.02 | 69.27 | 68.58 | 53.98 |
| LOLSGD + $\mathcal{L}_{rank}$ ❄ | 58.57 | 43.86 | 72.59 | 64.53 | 75.06 | 68.92 | 70.13 | 66.04 | 61.58 | 44.10 | 80.82 | 74.02 | 69.79 | 60.26 |
| LOLSGD + $\mathcal{L}_{distill}$ ❄ | 57.44 | 46.35 | 75.24 | 67.38 | 76.33 | 70.75 | 69.07 | 65.77 | 60.68 | 45.58 | 81.56 | 75.28 | 70.05 | 61.85 |
| LOLSGD + $\mathcal{L}_{distill}$ + $\mathcal{L}_{rank}$ ❄ | 60.83 | 51.75 | 75.75 | 70.25 | 76.70 | 74.26 | 70.13 | 69.54 | 65.71 | 53.10 | 82.95 | 80.31 | 72.01 | 66.54 |
| Best $\Delta$Acc$_{-Naive\ Target}$ | 15.87 | 42.69 | 23.36 | 46.50 | 17.22 | 41.35 | 15.47 | 36.39 | 18.79 | 46.76 | 20.64 | 46.23 | 18.51 | 44.18 |
| Oracle | 79.32 | 82.46 | 90.96 | 91.50 | 84.57 | 84.95 | 78.13 | 80.59 | 79.85 | 75.52 | 91.92 | 92.04 | 84.13 | 84.51 |

local SGD ($\frac{1}{M}$ epoch) and we run the same total epochs, for a fair computation budget. **Please see the supplementary for the details of setup, hyperparameters, and more analyses.**

## 4.1 Domain Adaptation on Partial Target Data

We first use the Office-Home dataset [74] with ResNet-50 [28] to study the effectiveness of the methods in section 3. Office-Home is a popular domain adaptation benchmark consisting of 65 object categories from four domains (Art, Clipart, Real, and Product).

**Methods.** We consider baselines including directly predicting with the **Source** model and naively fine-tuning it for a **Target** model. As proposed in section 3, we explore methods that promote covariate shifts by learning the limited parameters in normalization layers with the backbone parameters frozen, like **batchnorms (BN)** [31, 46] and an extra **instance normalization (IN)** [72] layer. We include another baseline **LP-FT** [42] proposed for the OOD generalization of fine-tuning but not for missing classes in HT. To compare with our LOLSGD, we also include related techniques in domain generalization based on stochastic weight average (**SWA**) [33] and its extension **SWAD** [7].

**Comparison.** We highlight the following observations in Table 3:

- **Naive fine-tuning is disruptive:** the unseen performance drops drastically, ultimately degrading overall accuracy and confirming the challenges of HT for all pairs of domains.
- **Normalization layers may help:** maintaining unseen accuracy and improving the seen little.
- **Freezing the classifier during training is necessary:** we found that the frozen classifier is crucial when the feature extractor is adapted (Avg. $+21\%$ Unseen vs. Target), motivating our approach of preserving class relationships subsection 3.3.
- **$\mathcal{L}_{distill}$ and $\mathcal{L}_{rank}$ regularize forgetting further:** largely mitigating unseen performance drops.
- **LOLSGD is effective:** without any regularizers, the proposed local SGD optimization can outperform the strong generalized SGD baselines SWA and SWAD.
- **LOLSGD with regularizers performs best overall:** as we suggested in section 3, LOLSGD needs to be carefully regularized while it adapts the domain styles.

**Effects of Source Ensemble (SE).** We further post-process the target models learned in Table 3 by ensembling with the source model, as introduced in subsection 3.3. We also include WISE [79] that ensembles the model weights instead of predictions. Generally, we found SE may reclaim some unseen performance, but not necessarily improve overall, as summarized in Table 4.

*Overall, we show that HT is promising*, improving the overall performance without sacrificing the unseen accuracy (highlighted in red in Table 3). Notably, our proposed solutions are based on optimization and agnostic to architectures. We will mainly focus on them later as they perform better.

Table 5: FEMNIST mean accuracy of 10 new writers.

| Methods | Overall | Seen | Seen (Chopping) | Unseen |
|---|---|---|---|---|
| Source | 84.67 | 88.67 | 89.60 | 64.99 |
| Naive Target | 82.67 | 92.07 | 92.07 | 35.60 |
| SGD✳ | 87.85 | 92.87 | 93.09 | 64.00 |
| SGD + $\mathcal{L}_{rank}$ ✳ | 88.18 | 93.29 | 93.51 | 64.00 |
| SGD + $\mathcal{L}_{distill}$ ✳ | 87.44 | 92.46 | 92.81 | 61.91 |
| LOLSGD✳ | 87.47 | 91.87 | 92.23 | 66.27 |
| LOLSGD + $\mathcal{L}_{rank}$ ✳ | 87.16 | 91.61 | 91.97 | 66.27 |
| LOLSGD + $\mathcal{L}_{distill}$ ✳ | 85.76 | 90.34 | 91.25 | 63.56 |
| LOLSGD + $\mathcal{L}_{distill}$ +SE✳ | 85.10 | 89.52 | 90.58 | 63.56 |

Table 6: iWildCam mean accuracy of 21 new locations.

| Methods | Overall | Seen | Seen (Chopping) | Unseen |
|---|---|---|---|---|
| Source | 35.71 | 35.74 | 58.40 | 26.43 |
| Naive Target | 35.38 | 51.08 | 52.72 | 1.90 |
| SGD✳ | 36.53 | 49.00 | 50.93 | 7.91 |
| SGD + $\mathcal{L}_{rank}$ ✳ | 36.28 | 40.98 | 56.13 | 19.76 |
| SGD + $\mathcal{L}_{distill}$ ✳ | 41.59 | 50.40 | 54.23 | 17.90 |
| LOLSGD✳ | 38.12 | 48.91 | 52.49 | 13.30 |
| LOLSGD + $\mathcal{L}_{rank}$ ✳ | 33.94 | 37.62 | 53.40 | 19.42 |
| LOLSGD + $\mathcal{L}_{distill}$ ✳ | 40.49 | 47.30 | 55.27 | 25.16 |
| LOLSGD + $\mathcal{L}_{distill}$ +SE✳ | 39.96 | 44.79 | 58.22 | 25.66 |

Table 7: Fine-tuning CLIP ViT-B/32 on VTAB. $50\%$ of classes each task are missing during training.

| Overall/Unseen Acc.  Methods | Caltech101 All. | Uns. | CIFAR100 All. | Uns. | DTD All. | Uns. | EuroSAT All. | Uns. | Flowers102 All. | Uns. | Pets All. | Uns. | Resisc45 All. | Uns. | SVHN All. | Uns. | SUN397 All. | Uns. | Avg. All. | Uns. |
|---|---|---|---|---|---|---|---|---|---|---|---|---|---|---|---|---|---|---|---|---|
| Source | 78.7 | 70.5 | 64.2 | 62.4 | 43.1 | 47.0 | 32.2 | 20.3 | 63.8 | 68.4 | 84.1 | 88.7 | 54.0 | 52.7 | 8.8 | 11.3 | 46.7 | 49.1 | 52.8 | 52.3 |
| Naive Target | 82.3 | 69.1 | 65.9 | 42.7 | 45.3 | 15.8 | 52.2 | 1.2 | 64.2 | 49.2 | 84.5 | 75.8 | 58.5 | 23.8 | 39.8 | 0.2 | 49.7 | 35.4 | 60.3 | 34.8 |
| SGD✳ | 82.3 | 69.0 | 66.1 | 42.7 | 45.5 | 16.4 | 52.5 | 1.8 | 64.5 | 49.8 | 84.7 | 75.8 | 57.9 | 21.9 | 40.1 | 0.5 | 49.8 | 36.0 | 60.4 | 34.9 |
| LOLSGD✳ | 82.6 | 69.8 | 69.6 | 51.8 | 46.0 | 18.8 | 52.7 | 2.7 | 65.3 | 51.0 | 85.9 | 78.2 | 59.4 | 27.1 | 36.7 | 0.7 | 52.5 | 45.5 | 61.2 | 38.4 |
| LOLSGD + $\mathcal{L}_{distill}$ ✳ | 82.0 | 69.4 | 67.1 | 49.1 | 48.0 | 22.9 | 52.7 | 3.8 | 66.3 | 52.9 | 83.8 | 74.2 | 60.0 | 29.0 | 35.1 | 0.5 | 52.6 | 47.6 | 60.8 | 38.8 |
| LOLSGD + $\mathcal{L}_{rank}$ ✳ | 82.8 | 70.3 | 70.0 | 52.5 | 45.3 | 18.4 | 51.4 | 1.4 | 64.2 | 50.9 | 84.6 | 76.8 | 59.5 | 27.6 | 34.5 | 0.3 | 52.8 | 46.1 | 60.6 | 38.3 |
| LOLSGD + $\mathcal{L}_{rank}$ +SE✳ | 82.5 | 70.8 | 72.3 | 58.4 | 52.6 | 37.3 | 54.5 | 8.4 | 69.2 | 62.5 | 87.3 | 83.2 | 64.8 | 39.7 | 35.8 | 3.8 | 53.4 | 50.0 | 63.6 | 46.0 |
| Oracle | 89.0 | 87.2 | 83.1 | 82.6 | 60.0 | 61.3 | 97.7 | 97.1 | 75.5 | 80.3 | 90.4 | 91.7 | 90.1 | 90.3 | 94.2 | 95.6 | 58.8 | 60.0 | 82.1 | 82.9 |

## 4.2 Realistic Holistic Transfer: FEMNIST and iWildCam Datasets

We simulate two realistic scenarios in which a new user collects a limited-size adaptation set and aims to customize a source model. Each new user has its natural domain shift.

**FEMNIST [4]**: a $62$-class hand-written characters dataset that consists of many writers [19] of different styles. We train a LeNet [44] on $40$ writers and adapt to each of the $10$ new writers. Each writer only has a few samples per class, thus when the data are split into training and test sets (in ratio of $7:3$) for personalization may suffer mismatched $P_{\mathcal{T}}(y)$.

**iWildCam [38, 2]**: animal recognition dataset of 181 classes of camera traps located worldwide. We train an ImageNet pre-trained ResNet-50 on 53 locations and adapt to 21 new locations. The data of each location are split by a timestamp; we train on previous data and test on future data.

Encouragingly, the observations on both natural datasets are generally consistent with the Office-Home experimental dataset. As shown in Table 5 and Table 6. The seen and unseen accuracy can be improved upon the source model at the same time, even without an ensemble. Here, we note that the seen class accuracy might be misleading. It might be two reasons if the seen class accuracy is high: 1) it adapts well to the target style and/or 2) the model simply collapses to an easier, fewer-way classifier. Just for a better understanding of the quality of the adapted features, we also report the seen classes' accuracy but "chopping out" the unseen classifiers in testing[2]. The seen accuracy with a chopped classifier is also competitive to naive fine-tuning, validating the feature quality is good; lower accuracy is mainly due to more classes making the problem harder.

## 4.3 Fine-Tuning on Zero-Shot CLIP ViT for Vision Tasks

Next, we go beyond domain adaptation and consider distribution shifts at the task level using the VTAB [92] benchmark that contains various vision tasks. We split each task by its labels; roughly half of the classes are considered unseen. We use the CLIP pre-trained ViT-B/32 [60]. Only those tasks in which the class names are in natural languages are considered using their text embeddings for zero-shot predictions. The results are summarized in Table 7. We further compare to the CoCoOp [96] fine-tuning method designed for handling unseen classes in the supplementary.

Across the tasks, we see HT methods remain effective in most cases. Interestingly, we see some exceptions that the degrees of improvement become diverged, which inspires our discussions next.

---

[2]In realistic cases, the fully-observed environments are not available for the oracle upper bounds.

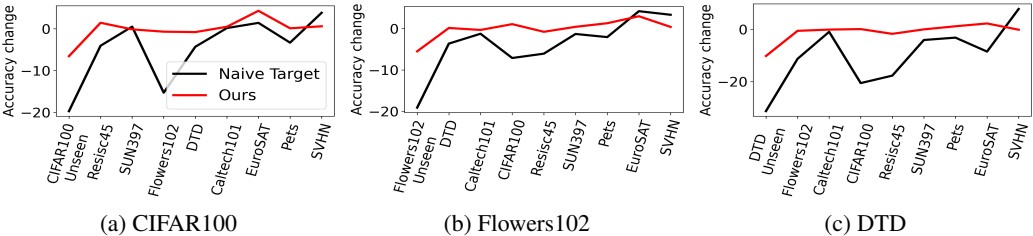

|               |               |               |
| ------------- | ------------- | ------------- |
| (a) CIFAR100  | (b) Flowers102 | (c) DTD      |

Figure 5: **Disparate impact: will fine-tuning improve similar tasks?** We fine-tune CLIP on the training set of a task and evaluate on others. Naively fine-tuned target models actually *degrade more* on more similar tasks, while our HT method is much more robust. Tasks are sorted in the x-axis based on Affinity Score following [25].

## 4.4 Studies: When should HT Work and Not Work?

**Semantic gaps of the source model.** As discussed in subsection 2.3, HT relies on the DA assumption that the source and target tasks share similar semantics, i.e., $P_\mathcal{S}(y|\boldsymbol{x}) \approx P_\mathcal{T}(y|\boldsymbol{x})$. There should be little semantic shifts from the source to the target. In Table 7, we found the unseen class performance of some tasks is very low, and applying HT methods cannot help much. We hypothesize that the images are likely to be predicted as other meanings by the CLIP model rather than the goal of the target task. Looking at the Street View House Numbers (SVHN) dataset [54], we adversarially add class names like "number plate" to confuse with the digits (e.g., "9"). Interestingly, we found about 60% of the cases, CLIP fails to predict a number (Table 8). This validates that $P_\mathcal{S}(y|\boldsymbol{x})$ pre-trained by CLIP is quite different from the digit classification, therefore, it becomes very hard if not impossible for it to recognize unseen digits correctly. Similarly, DTD (texture) and EuroSAT (land use) could suffer from the same concept shift issue. Resolving it requires a "task-level generalization" (e.g., object recognition → digits classification), which is obviously out of the scope of HT.

**Disparate impact.** If we fine-tune a task, should it benefit other similar tasks? We reveal that the answer might be *negative* in HT settings. We fine-tune CLIP on a task following Table 7, evaluate on *others* using the corresponding CLIP text classifiers, and summarize the accuracy difference to the original CLIP in Figure 5. Intriguingly, more similar tasks in fact drop more, likely because they are *disrupted* by the fine-tuned task most. This again highlights the importance of HT; we found our method makes it more robust.

**Case study: fine-tuning can lead to serious false negatives.** So far, we mainly evaluate HT from the lens of unseen class accuracy. Here, we discuss confusion in predictions and a case that transfers without considering HT could pose even larger risks. Considering training a classifier for recognizing mushrooms on some non-toxic ones and encountering a visually similar but toxic one in testing, the classifier will likely label it as one of the non-toxic ones. Even worse, such bias is exaggerated by the adaptation, due to the partial training classes. We build an extreme case that selects 6 pairs of fungi (toxic/non-toxic each) from the iNaturalist dataset [73] and fine-tunes on only the non-toxic ones from CLIP. The results in Table 9 show such concern exists and should be considered in future algorithm designs.

Table 8: Examinizing HT of zero-shot CLIP on SVHN dataset.

| Classes          | Predictions |
| ---------------- | ----------- |
| "0" to "9"       | 39.4%       |
| "number plate"   | 32.7%       |
| "house number"   | 16.8%       |
| "wallpaper"      | 11.1%       |
| 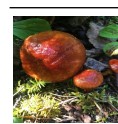    |    |
| Missing rate     | 60.6%       |

Table 9: Fine-tuning can lead to serious false negatives.

| Method  | Seen Accuracy | False Negatives |
| ------- | ------------- | --------------- |
| Source  | 23.3%         | 46.7%           |
| Target  | 63.3%         | 96.7%           |
| Our HT  | 60.0%         | 83.3%           |

| Toxic | Non-Toxic (Train) |
| ----- | ----------------- |
| 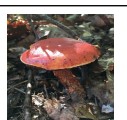 |         |

## 5 Conclusions (Related Works & Limitations in Supplementary)

We introduce a novel and practical transfer learning problem that emphasizes generalization to unseen classes in the target domain but seen in the source domain. Despite its challenges, we establish strong baselines and demonstrate the potential for improving both seen and unseen target classes simultaneously, paving the way for holistic transfer. To facilitate further research, we construct a comprehensive benchmark with diverse scenarios and conduct insightful experiments. In future work, we envision exploring various directions, including improved disentanglement of domain styles and classes, integrating our approach with other paradigms like test-time adaptation, and extending beyond classification tasks.

## Acknowledgment

This research is supported by grants from the National Science Foundation (OAC-2118240, OAC-2112606). We are thankful for the generous support of the computational resources by the Ohio Supercomputer Center.

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

# Supplementary Material

We provide details omitted in the main paper.

- Appendix A: related work (cf. subsection 2.3 and section 5 of the main paper).
- Appendix B: additional benchmark details (cf. subsection 2.2 of the main paper).
- Appendix C: additional training details (cf. section 4 of the main paper).
- Appendix D: additional results and analyses (cf. section 4 of the main paper).
- Appendix E: additional discussions (cf. section 5 of the main paper).

## A Related Work

We review related work on other transfer learning paradigms. We briefly describe their settings and distinguish their differences from our proposed holistic transfer (HT) problem.

### A.1 Domain Adaptation

Domain adaptation (DA) is the most iconical machine learning setting to tackle the domain-shift problem [86, 27, 62, 65, 14, 49]. With the common objective of transferring source-domain knowledge to target domains, various settings have been proposed to incorporate different constraints and assumptions. The assumption can be the degrees of overlap between the source and the target label sets [90, 64, 5, 6, 85, 57, 34, 9]. To relax the constraint of accessing source data, source-free DA can solely rely on the target data for adaptation [87, 23, 43, 18]. Despite the abundant variations, DA settings all share one common assumption: the target distributions in training and testing are matched, making our HT fundamentally different from them. In our HT, we can encounter target test classes that are unseen in the target training set but seen in the source domain. Therefore, HT requires a distinct ability that can generalize the style shifts learned on the target seen classes to other unseen classes.

### A.2 Out-of-domain Generalization

Although fine-tuning a pre-trained model often leads to impressive accuracy for downstream tasks, recent studies have revealed that it may compromise the out-of-domain (OOD) robustness of the model [41, 3, 79]. Several robust fine-tuning methods are thus proposed to balance the trade-off between in-domain downstream accuracy and OOD generalization [70, 61, 82]. LP-FT [41] proposed to learn a classifier with frozen features before end-to-end fine-tuning to avoid feature distortion. Some other approaches relied on ensembles with pre-trained models to increase the robustness [78, 30]. However, the main focus of these studies remains on preserving the robustness to different input styles for classes seen in the target training set. This is significantly different from HT. Our HT problem aims to generalize the styles for classes unseen in the target training set.

### A.3 Continal Learning

The goal of continual learning (CL) is to sequentially adapt to multiple tasks without catastrophically forgetting the previously learned ones [37, 51, 47]. To achieve this goal, existing studies have proposed to exploit a replay buffer for storing old data [63, 8, 80, 71, 67, 88, 50, 97, 52, 95], or to constrain the fine-tuning with old models [89, 22, 1, 24, 77]. Unlike HT, CL still assumes all the encountered training distributions, which could be many, are aligned with their corresponding test distributions. Although reducing forgetting can be the first step for HT to maintain unseen class accuracy, we argue that this is insufficient in HT due to the source-target domain mismatch. Adapting the features for unseen classes to the target domain remains a key challenge for HT. Moreover, HT can also be potentially compatible with CL to consider learning on a non-iid data stream.

### A.4 Zero-shot Learning

Zero-shot learning tackles the setting where training and test classes are completely disjoint [81, 16, 84, 58]. As no training data are available for test classes, the main challenge resides in learning source

Table 10: A summary of the dataset statistics for our HT benchmark.

| Datasets | Source domains | Target domain | #Classes | #Seen classes | #Target training | #Target test |
|---|---|---|---|---|---|---|
| Office-Home | Art | Clipart | 65 | 30 | 1,471 | 1,330 |
| | | Product | | | 1,265 | 1,361 |
| | | Real | | | 1,413 | 1,335 |
| | Real | Art | 65 | 30 | 857 | 750 |
| | | Clipart | | | 1,493 | 1,330 |
| | | Product | | | 1,459 | 1,361 |
| FEMNIST | 40 writers | 10 new writers | 62 | Vary by data collection bias | Vary by data collection bias | Vary by data collection bias |
| iWildCam | 53 camera trap locations | 21 new camera trap locations | 181 | Vary by data collection bias | Vary by data collection bias | Vary by data collection bias |
| VTAB | CLIP | Caltech101 | 102 | 51 | 1,371 | 6,084 |
| | | CIFAR100 | 100 | 50 | 22,513 | 10,000 |
| | | DTD | 47 | 23 | 920 | 1,880 |
| | | EuroSAT | 10 | 5 | 8,424 | 5,400 |
| | | Flowers102 | 102 | 51 | 510 | 6,149 |
| | | Pets | 37 | 18 | 1,445 | 3,669 |
| | | Resisc45 | 45 | 22 | 9,159 | 6,300 |
| | | SVHN | 10 | 5 | 28,197 | 26,032 |
| | | SUN397 | 397 | 198 | 37,542 | 21,750 |
| iNaturalist (Fungi) | CLIP | Fungi | 12 | 6 | 30 | 60 |

features that can generalize to unseen semantic meanings. To achieve this, auxiliary information (e.g., texts or attributes) is usually needed to describe the test classes and connect them back to the training classes [83, 15, 53, 45]. In HT, we assume the missing classes in target domains are already seen in the source domain. We make this assumption to simplify the problem so that HT can focus on generalizing the domain shifts to unseen classes. However, we argue that HT is compatible with zero-shot learning to make the setting more flexible.

# B    Additional Benchmark Details

To support the study of the HT problem, we create a benchmark that covers extensive scenarios across both experimental and realistic public datasets. We provide details about these datasets.

## B.1    Office-Home

**Setup.** We consider the standard domain adaptation setting but with some missing classes in the target training sets. We use the popular Office-Home dataset consisting of 65 categories from 4 domains (Art, Clipart, Real, and Product). In our benchmark, we use Art and Real as source domains; each source domain is then transferred to each of the three remaining target domains individually, resulting in six source-target pairs. For each source-target pair, we use all the data in the source domain to train a source model. Then, for each target domain, we randomly split the data of each class into training and test sets with a ratio of 7:3. We randomly sample 30 seen classes and combine the training data of these seen classes to create the target training set. Finally, the target test set consists of the test images of all 65 classes in the target domain. A summary of the statistics can be found in Table 10.

**Evaluation.** We follow the standard evaluation metric in the Office-Home dataset to compute the overall accuracy for each source-target pair. Besides, we explicitly compute the accuracy of the unseen-class data to evaluate the transferring performance of the unseen classes. The average accuracy over all the source-target pairs is also reported.

## B.2    FEMNIST

**Setup.** The FEMNIST dataset contains 62-class hand-written characters from many writers with different writing styles. In practical personalized AI applications, it is common to train on data from many users but needs to adapt to many new users after the (source) model is trained [11]. As we

can only collect a limited-size data set for each writer, each writer's data only cover a subset of the 62-class characters, resulting in the need for HT. We randomly sample 40 writers whose data combined can cover all 62 classes and use their data to train a source model. Then, we randomly sample 10 new writers. Each new writer's data is divided into training and test sets in a ratio of 7:3. Note that each client may not have enough images per class, which creates a realistic scenario of personalization with limited samples, which results in a mismatch of the class distributions between training and test sets. The dataset statistics are summarized in Table 10.

**Evaluation.** We report the overall accuracy averaged over all the 10 new writers. To evaluate the trade-off between seen and unseen classes, we also report the averaged accuracy on the seen and unseen classes, respectively. As this dataset has no oracle training set for each new writer, we report the seen accuracy computed by chopping out unseen classes in the classifier to evaluate the quality of the adapted features.

### B.3   iWildCam

**Setup.** We consider a realistic scenario of HT, where we initially have abundant camera traps installed across many geo-locations (source domains) and now need to transfer to a new camera trap location (target domain). In the new location, we can only use the data collected within a fixed amount of time in the beginning (e.g., the first month) as our target training set. As it is impossible for all the animal species to appear in the first month, the target training data can be biased toward some classes that show up. This is a natural data collection bias caused by time.

We start from the iWildCam dataset in the WILDS [38] benchmark. As we mainly focus on animal species classification, we remove the "empty" class for simplicity and thus obtain a total of 181 classes. For each camera trap location, we sort the images by their timestamps and group images into sequences if the difference in their timestamps is smaller than 30 minutes, to avoid information leaks. We randomly sample 53 camera trap locations whose images cover all 181 classes and use all their data to train a source model. For each of the remaining locations, we randomly sample training and test sets based on a ratio of 7:3. We only keep locations with more than 500 images in both the training and test sets, thereby resulting in 21 new locations for adaptation. For each new location, we form the target training set by sorting the training images by time and only using the first 25% of them. A summary of the dataset statistics is given in Table 10.

**Evaluation.** We report the overall accuracy averaged over all 21 new locations. To evaluate the trade-off between seen and unseen classes, we also report the averaged accuracy on the seen and unseen classes, respectively. As this dataset has no oracle training set for each new location, we report the seen accuracy computed by chopping out unseen classes in the classifier to evaluate the quality of the adapted features.

### B.4   VTAB

**Setup.** We consider another practical use of HT by going beyond domain adaptation and fine-tuning the zero-shot CLIP [60] for distribution shifts at the task levels. We use the VTAB [92] benchmark that includes various image classification tasks. To enable zero-shot predictions, we only use the tasks that provide text names for classes, thereby resulting in 9 tasks: Caltech101, CIFAR100, DTD, EuroSAT, Flowers102, Pets, Resisc45, SVHN, and SUN397. We use the standard training and test sets provided by the VTAB benchmark. Then, we randomly sample half of the classes as seen and the remaining as unseen. The target training set only includes the training images of the seen classes, while the target test set contains all the test images. A summary of the statistics of this dataset is shown in Table 10.

**Evaluation.** Following the standard evaluation in VTAB, we report the overall accuracy for each of the 9 tasks. Besides, we also compute the accuracy on the unseen-class data to evaluate the transferring performance of unseen classes. Finally, the average accuracy across all 9 tasks is also reported.

### B.5   iNaturalist (2021 Version, Fungi)

**Setup.** To demonstrate the impact of visually similar classes in HT, we carefully pick 6 pairs of fungi classes from the iNaturalist dataset, thus resulting in a total of 12 classes. Each pair of fungi classes corresponds to 2 species of visually similar fungi; one is non-toxic, while the other one is toxic. We

use the zero-shot CLIP model with the fungi names as our source model. Then, the training images from the 6 non-toxic fungi classes form the target training set. The target test set consists of all the test images from all 12 classes. A summary of the dataset statistics and some examples are shown in Table 10.

**Evaluation.** We report the seen accuracy on the target test set to evaluate the adaptation performance. As wrongly predicting toxic fungi as non-toxic ones can result in severe outcomes, we also report the false negative rate, which is computed as the percentage of the images of toxic fungi being predicted as non-toxic fungi classes.

## C   Additional Training Details

We provide the training details for our results reported in section 4.

For the Office-Home dataset, we initialize a ResNet-50 with ImageNet pre-trained weights. Then, we train it on the source domain for 20 epochs using the SGD optimizer with a learning rate 1e-3, momentum 0.9, weight decay 5e-4, and batch size 64. For all methods that adapt to the target domains, we fine-tune the source model for 20 epochs using the SGD optimizer with a learning rate 1e-4, momentum 0.9, weight decay 5e-4, and batch size 64. For our suggested HT methods, we set the hyper-parameters $\mathcal{L}_{\text{distill}} = 10$ and $\mathcal{L}_{\text{rank}} = 100$.

For the FEMNIST dataset, we train a LeNet from scratch on the data of the 40 source writers for 100 epochs using the SGD optimizer with a learning rate 1e-2, momentum 0.9, weight decay 5e-4, and batch size 32. To adapt to each new writer, we fine-tune the source model for 10 epochs using the SGD optimizer with a learning rate 1e-3, momentum 0.9, weight decay 1e-4, and batch size 32. We set the hyper-parameters $\mathcal{L}_{\text{distill}} = 0.1$ and $\mathcal{L}_{\text{rank}} = 10$.

For the iWildCam dataset, we train a ResNet-50, which is initialized with ImageNet pre-trained weights, on the data of source camera trap locations for 50 epochs using the SGD optimizer with a learning rate 3e-5, momentum 0.9, weight decay 0.0, and batch size 16. When adapting to each new location, we fine-tune the source model for 20 epochs using the SGD optimizer with a learning rate 3e-6, momentum 0.9, weight decay 0.0, and batch size 16. We set the hyper-parameters $\mathcal{L}_{\text{distill}} = 50$ and $\mathcal{L}_{\text{rank}} = 200$.

For the VTAB benchmark, we use the class names for each of the 9 tasks to form the zero-shot CLIP models, which are ViT-B/32. We fine-tune the source model on target tasks for 20 epochs using the SGD optimizer with a learning rate 1e-5, momentum 0.9, weight decay 0.0, and batch size 64. We set the hyper-parameters $\mathcal{L}_{\text{distill}} = 1$ and $\mathcal{L}_{\text{rank}} = 5$.

For the iNaturalist Fungi dataset, we use the fungi species names to build a zero-shot CLIP model with a ViT-B/32 architecture. We then fine-tune the source model on the target training set for 5 epochs using the SGD optimizer with a learning rate 5e-5, momentum 0.9, weight decay 0.0, and batch size 5. We set the hyper-parameters $\mathcal{L}_{\text{distill}} = 1$ and $\mathcal{L}_{\text{rank}} = 1$.

## D   Additinal Results and Analyses

### D.1   Variances of the Results in section 4

We provide variances of our results reported in our main paper. We compute the variances across 3 random seeds. Table 11 shows the variances of the test accuracy on Office-Home. The variances of the mean accuracy on FEMNIST and iWildCam are provided in Table 12 and in Table 13, respectively. Finally, Table 14 gives the variances of the test accuracy for each of the 9 tasks in VTAB. These results reveal that the reported accuracy is relatively robust across random seeds.

### D.2   More Discussions on Seen and Unseen Classes

**Different numbers of images per seen class.** In the real world, it is unrealistic for end-users to collect data for all classes before adaptation. To further consider a lower data collection cost, we reduce the number of training images per seen class to study its effects. We conduct the experiment on the Office-Home dataset with "Art" as our source domain and "Clipart" as our target domain. Specifically, we randomly sample 10% and 50% of the training images for each seen class and

Table 11: Varainces of domain adaptation 65-way test accuracy on Office-Home with 30 seen and 35 unseen classes (cf. Table 3). We compute the variances over 3 random seeds. Blue: HT methods suggested by us in section 3. Red: methods that significantly improve overall accuracy and successfully maintain unseen accuracy on the source model. ❄: the linear classifier is frozen during training.

| Domains: source→target Methods / Acc. | Ar→Cl Overall | Ar→Cl Unseen | Ar→Pr Overall | Ar→Pr Unseen | Ar→Rw Overall | Ar→Rw Unseen | Rw→Ar Overall | Rw→Ar Unseen | Rw→Cl Overall | Rw→Cl Unseen | Rw→Pr Overall | Rw→Pr Unseen | Avg. Overall | Avg. Unseen |
|---|---|---|---|---|---|---|---|---|---|---|---|---|---|---|
| Source | 0.64 | 0.11 | 1.30 | 1.35 | 0.25 | 0.24 | 0.82 | 0.90 | 0.23 | 1.40 | 0.30 | 0.49 | 0.05 | 0.11 |
| Naive Target | 0.02 | 0.01 | 0.14 | 0.02 | 0.27 | 0.53 | 0.93 | 1.19 | 0.00 | 0.36 | 0.35 | 1.11 | 0.04 | 0.09 |
| BN only | 1.11 | 3.11 | 0.22 | 0.27 | 0.41 | 1.07 | 1.71 | 7.58 | 0.68 | 2.44 | 1.43 | 4.47 | 0.03 | 0.17 |
| BN (stats) only | 0.63 | 0.06 | 1.00 | 1.63 | 0.02 | 0.32 | 0.08 | 0.17 | 0.16 | 0.31 | 0.15 | 0.05 | 0.14 | 0.16 |
| BN (stats) only | 0.57 | 0.05 | 0.87 | 0.58 | 0.25 | 0.18 | 0.05 | 0.02 | 0.21 | 0.02 | 0.30 | 0.37 | 0.12 | 0.03 |
| LP-FT | 0.14 | 0.14 | 0.05 | 0.04 | 0.08 | 0.18 | 0.16 | 0.61 | 0.01 | 0.31 | 0.35 | 0.87 | 0.02 | 0.04 |
| SGD (w/ frozen classifier ❄) | 0.02 | 0.52 | 0.63 | 1.27 | 0.75 | 0.98 | 0.07 | 0.31 | 0.57 | 0.85 | 0.09 | 0.60 | 0.07 | 0.18 |
| SGD + $\mathcal{L}_{\text{distill}}$ ❄ | 0.28 | 0.09 | 0.03 | 0.01 | 0.70 | 0.73 | 0.23 | 0.94 | 0.05 | 0.09 | 0.15 | 1.02 | 0.02 | 0.01 |
| SGD + $\mathcal{L}_{\text{rank}}$ ❄ | 1.02 | 0.78 | 0.39 | 0.48 | 0.88 | 1.13 | 0.01 | 0.51 | 0.07 | 0.67 | 0.09 | 0.03 | 0.08 | 0.05 |
| SWA ❄ | 0.23 | 0.73 | 0.62 | 1.44 | 0.33 | 0.60 | 0.45 | 0.31 | 1.39 | 3.39 | 0.13 | 0.65 | 0.02 | 0.15 |
| SWAD ❄ | 0.43 | 0.83 | 0.62 | 1.79 | 0.35 | 0.42 | 0.45 | 0.32 | 1.64 | 3.71 | 0.08 | 0.44 | 0.03 | 0.17 |
| LOLSGD❄ | 0.36 | 0.20 | 0.62 | 1.00 | 0.20 | 1.25 | 0.73 | 0.61 | 1.20 | 1.59 | 0.17 | 0.65 | 0.00 | 0.11 |
| LOLSGD +$\mathcal{L}_{\text{rank}}$ ❄ | 0.02 | 0.05 | 0.24 | 0.95 | 0.06 | 0.05 | 0.34 | 0.22 | 0.63 | 1.09 | 0.16 | 0.96 | 0.04 | 0.14 |
| LOLSGD +$\mathcal{L}_{\text{distill}}$ ❄ | 0.14 | 0.56 | 0.79 | 0.54 | 0.25 | 0.09 | 0.22 | 0.17 | 0.28 | 1.88 | 0.07 | 0.08 | 0.06 | 0.01 |
| LOLSGD +$\mathcal{L}_{\text{distill}}$ + $\mathcal{L}_{\text{rank}}$ ❄ | 0.18 | 0.06 | 0.75 | 1.82 | 0.00 | 0.34 | 0.26 | 0.65 | 0.75 | 0.36 | 0.01 | 0.23 | 0.12 | 0.09 |
| Oracle | 0.05 | 0.05 | 0.06 | 0.16 | 0.01 | 0.16 | 0.17 | 0.02 | 0.22 | 0.05 | 0.35 | 0.20 | 0.02 | 0.00 |

Table 12: Varainces of FEMNIST mean accuracy of 10 new writers (cf. Table 5). We compute the variances over 3 random seeds.

| Methods | Overall | Seen | Seen (Chopping) | Unseen |
|---|---|---|---|---|
| Source | 0.44 | 0.12 | 0.07 | 5.12 |
| Naive Target | 0.49 | 0.77 | 0.77 | 5.69 |
| SGD❄ | 0.43 | 0.17 | 0.11 | 1.90 |
| SGD + $\mathcal{L}_{\text{rank}}$ ❄ | 0.55 | 0.08 | 0.04 | 4.05 |
| SGD + $\mathcal{L}_{\text{distill}}$ ❄ | 0.37 | 0.19 | 0.13 | 1.72 |
| LOLSGD❄ | 0.48 | 0.10 | 0.17 | 4.73 |
| LOLSGD +$\mathcal{L}_{\text{rank}}$ ❄ | 0.37 | 0.10 | 0.20 | 2.04 |
| LOLSGD +$\mathcal{L}_{\text{distill}}$ ❄ | 0.48 | 0.16 | 0.37 | 4.90 |
| LOLSGD +$\mathcal{L}_{\text{distill}}$ +SE❄ | 0.65 | 0.26 | 0.06 | 6.23 |

Table 13: Variances of iWildCam mean accuracy of 21 new locations (cf. Table 6). We compute the variances over 3 random seeds.

| Methods | Overall | Seen | Seen (Chopping) | Unseen |
|---|---|---|---|---|
| Source | 0.12 | 1.45 | 5.29 | 0.86 |
| Naive Target | 1.66 | 3.85 | 4.45 | 0.04 |
| SGD❄ | 0.10 | 1.48 | 1.67 | 0.01 |
| SGD + $\mathcal{L}_{\text{rank}}$ ❄ | 0.94 | 7.73 | 2.41 | 2.35 |
| SGD + $\mathcal{L}_{\text{distill}}$ ❄ | 3.76 | 1.26 | 1.32 | 4.07 |
| LOLSGD❄ | 0.79 | 0.37 | 0.76 | 1.63 |
| LOLSGD +$\mathcal{L}_{\text{rank}}$ ❄ | 1.70 | 7.85 | 5.03 | 3.98 |
| LOLSGD +$\mathcal{L}_{\text{distill}}$ ❄ | 1.33 | 0.67 | 1.86 | 3.36 |
| LOLSGD +$\mathcal{L}_{\text{distill}}$ +SE❄ | 1.17 | 1.07 | 2.58 | 0.15 |

fine-tune the source model for the *same iterations* for fair comparisons. Interestingly, Table 15 shows that naive fine-tuning can obtain higher unseen accuracy, compared to naive fine-tuning with more data. The reason might be that training with more data needs to update the model weights more, making the unseen classes easier to forget. In contrast, applying our suggested HT methods, especially for LOLSGD with our regularization, the unseen classes can be better maintained across different training data sizes.

**Unseen performance vs. overall performance.** One might observe that sometimes the unseen performance could be better than the overall performance. We surmise this is because the classes are not equally difficult for classification. In some cases, before we perform adaptation, we can already see that the unseen class accuracy of the source model is higher than its overall accuracy, meaning that those unseen classes are inherently easier than the seen classes. Some methods (e.g., BN (stats) only) can better keep the unseen accuracy close to the source model with less forgetting. However, they cannot adapt the seen classes effectively to the target domain. Therefore, the results of these methods generally follow the trend in the source model with the unseen accuracy higher than the overall accuracy. In contrast, some methods (e.g., LP-FT) can better adapt the seen classes to the target domain but suffer from serious forgetting of the unseen classes. These methods thus have lower unseen accuracy.

**New classes in the target domain.** We note that, in our current setup all the experiments assume the classes tested in the target domain appeared in source training. If some "seen" classes in the target domain are not in the source domain, this will require expanding the label space of the model. A simple baseline would be first training the classification weights for those "seen" classes from scratch while keeping all other model components intact. Then, we can apply our holistic transfer approach to the expanded model. A more sophisticated solution would involve techniques from continual

Table 14: Variances of test accuracy for fine-tuning CLIP ViT-B/32 on VTAB (cf. Table 7). $50\%$ of classes in each task are missing during training. We compute the variances over 3 random seeds.

| Overall/Unseen Acc. Methods | Caltech101 All | Caltech101 Uns. | CIFAR100 All | CIFAR100 Uns. | DTD All | DTD Uns. | EuroSAT All | EuroSAT Uns. | Flowers102 All | Flowers102 Uns. | Pets All | Pets Uns. | Resisc45 All | Resisc45 Uns. | SVHN All | SVHN Uns. | SUN397 All | SUN397 Uns. | Avg. All | Avg. Uns. |
|---|---|---|---|---|---|---|---|---|---|---|---|---|---|---|---|---|---|---|---|---|
| Source | 0.00 | 0.00 | 0.00 | 0.00 | 0.00 | 0.00 | 0.00 | 0.00 | 0.00 | 0.00 | 0.00 | 0.00 | 0.00 | 0.00 | 0.00 | 0.00 | 0.00 | 0.00 | 0.00 | 0.00 |
| Naive Target | 0.00 | 0.00 | 0.02 | 0.08 | 0.17 | 1.32 | 0.01 | 0.02 | 0.30 | 0.45 | 0.05 | 0.38 | 0.22 | 1.45 | 0.01 | 0.03 | 0.01 | 0.05 | 0.00 | 0.01 |
| SGD ❄ | 0.00 | 0.01 | 0.02 | 0.17 | 0.31 | 0.33 | 0.21 | 0.59 | 0.16 | 0.07 | 0.03 | 0.15 | 0.13 | 1.00 | 0.03 | 0.02 | 0.00 | 0.01 | 0.02 | 0.02 |
| LOLSGD ❄ | 0.01 | 0.00 | 0.01 | 0.03 | 0.39 | 1.33 | 0.16 | 0.69 | 0.21 | 0.12 | 0.03 | 0.13 | 0.01 | 0.07 | 2.29 | 0.14 | 0.06 | 0.29 | 0.04 | 0.01 |
| LOLSGD $+\mathcal{L}_{\text{distill}}$ ❄ | 0.00 | 0.00 | 0.01 | 0.04 | 0.04 | 0.00 | 0.13 | 0.20 | 0.01 | 0.02 | 0.01 | 0.00 | 0.01 | 0.02 | 1.80 | 0.01 | 0.00 | 0.00 | 0.01 | 0.00 |
| LOLSGD $+\mathcal{L}_{\text{rank}}$ ❄ | 0.00 | 0.01 | 0.00 | 0.02 | 0.07 | 0.07 | 0.17 | 0.34 | 0.06 | 0.02 | 0.02 | 0.01 | 0.01 | 0.10 | 1.37 | 0.36 | 0.03 | 0.06 | 0.01 | 0.00 |
| LOLSGD $+\mathcal{L}_{\text{rank}}$ +SE ❄ | 0.01 | 0.02 | 0.07 | 0.14 | 0.12 | 0.33 | 0.06 | 0.05 | 0.06 | 0.03 | 0.00 | 0.03 | 0.05 | 0.09 | 1.91 | 0.82 | 0.01 | 0.05 | 0.00 | 0.02 |
| Oracle | 0.29 | 0.19 | 0.00 | 0.01 | 0.40 | 0.50 | 0.01 | 0.07 | 0.27 | 0.35 | 0.02 | 0.14 | 0.02 | 0.12 | 0.00 | 0.00 | 0.00 | 0.01 | 0.02 | 0.03 |

Table 15: Different percentages of the target training data for each seen class on Office-Home: Ar $\rightarrow$ Cl.

| % of target training Methods/Acc. | 10% Overall | 10% Unseen | 50% Overall | 50% Unseen | 100% Overall | 100% Unseen |
|---|---|---|---|---|---|---|
| Source | 47.07 | 50.29 | 47.07 | 50.29 | 47.07 | 50.29 |
| Naive Target | 41.73 | 19.74 | 43.46 | 9.94 | 44.96 | 9.06 |
| SGD ❄ | 51.28 | 41.96 | 52.33 | 28.36 | 52.11 | 24.12 |
| SGD $+ \mathcal{L}_{\text{rank}}$ ❄ | 50.45 | 46.49 | 56.02 | 40.20 | 59.17 | 39.47 |
| SGD $+ \mathcal{L}_{\text{distill}}$ ❄ | 48.80 | 41.37 | 53.91 | 38.30 | 56.54 | 39.18 |
| LOLSGD ❄ | **52.63** | 46.20 | 54.21 | 34.94 | 56.47 | 35.09 |
| LOLSGD $+\mathcal{L}_{\text{rank}}$ ❄ | 51.13 | 48.83 | 55.86 | 44.74 | 58.57 | 43.86 |
| LOLSGD $+\mathcal{L}_{\text{distill}}$ ❄ | 51.28 | 45.91 | 55.19 | 44.88 | 57.44 | 46.35 |
| LOLSGD $+\mathcal{L}_{\text{distill}} + \mathcal{L}_{\text{rank}}$ ❄ | 51.05 | **50.88** | **58.65** | **52.05** | **60.83** | **51.75** |

learning, or more specifically, class-incremental learning. This machine learning paradigm aims to expand the label space of a model. We leave a suitable combination of our approach and techniques from class-incremental learning as future work.

## D.3 Effects of the Source Ensemble Coefficients

In section 4, we apply Source Ensemble (with a mixing coefficient $\alpha = 0.5$) to reclaim some ability of the source model to maintain the unseen accuracy (cf. Table 4). To further understand the trade-off between the source and the fine-tuned target models, we study the effects of the mixing coefficient $\alpha$ by varying it between $[0, 1]$. We conduct our study on Office-Home and report the overall and unseen accuracy averaged over all the source-target domain pairs. As shown in Figure 6, applying either SE or WiSE cannot save the naively fine-tuned target model from being heavily biased to seen classes. On SGD with frozen classifiers, our SE shows a better trade-off than WISE [79]. Finally, fine-tuning target models with our suggested HT methods can clearly yield the best trade-off.

## D.4 Comparison to CoCoOp [96] on CLIP

In our experiments, we use CLIP to construct the source models for the VTAB and iNaturalist experiments. We further compared to a recent CLIP fine-tuning method CoCoOp [96] in these experiments. CoCoOp fine-tunes CLIP by training a meta-net to condition the classification weights (i.e., the fully-connected layer) on the input images. In other words, CoCoOp freezes the visual features but changes the classifier weights by minimizing the standard cross-entropy loss. In contrast, our approach freezes the classifier weights but adapts the visual features by minimizing the loss designed specifically for HT.

We conduct two experiments: 1) CoCoOp alone for the HT problem, and 2) combining CoCoOp with our approach. For 2), we take the resulting model after CoCoOp as the improved source model and further adapt the feature. We report the result of the CIFAR-100 task in VTAB (cf. Table 7). As shown in Table 16, CoCoOp alone performs well on unseen classes, but it improves the overall accuracy only marginally. Combining both approaches, we can obtain better accuracy in unseen and seen classes, leading to the best overall accuracy.

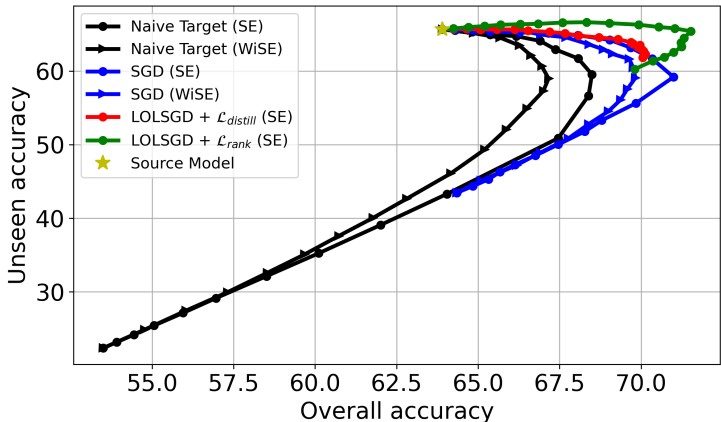

Figure 6: **Effects of Source Ensembles.** Ensemble the source (the star marker) and the target models (the end point of each line from the source model) with a mixing coefficient $\alpha \in [0, 1]$ on Office-Home.

## D.5 More experiments with the updating BN baseline

Batchnorm layers are critical components in feed-forward neural networks especially when learning with distributional shift [46, 94]. In cf. Table 3, one may observe the baseline "BN (stats only)" of updating only the means and variances in batchnorm layers performs surprisingly well on the unseen accuracy. We note that the ultimate goal of HT is to achieve high overall accuracy, not merely unseen accuracy. Although "BN (stats only)" maintains the unseen accuracy well on the Office-Home dataset, it cannot effectively improve the seen accuracy, resulting in worse overall accuracy. We apply "BN (stats only)" to iWildCAM (cf. Table 6). As in Table 17, it maintains the unseen accuracy well but cannot improve the seen accuracy. That said, we think that a more sophisticated, dedicatedly designed version of BN might perform better, and we leave it as future work.

## D.6 A study for LOLSGD canceling out class biases

Our LOLSGD aims to disentangle them or, more precisely, reduce the disruptive concept shift by subsampling multiple datasets that each contain a subset of the seen classes. When updating the model separately in parallel with these subsampled datasets, each updated model is affected by a different concept shift but a shared covariate shift. By averaging these models, LOLSGD can potentially cancel out the disruptive concept shifts and strengthen the shared covariate shift. We provide more evidence that LOLSGD can cancel out the disruptive concept shifts. We compare the seen class accuracy among 1) naive fine-tuning with the partial target data, 2) LOLSGD, and 3) fine-tuning with the full target data (i.e., the oracle model). As shown in Table 19, the seen class accuracy of naive fine-tuning (black dotted line) exceeds the oracle (green dotted line), indicating that naive fine-tuning learns undesired concept shifts towards seen classes, leading to an unreasonable accuracy. In contrast, the seen accuracy of LOLSGD (red dotted line) consistently stays below the oracle, indicating that undesired concept shifts are reduced. As a result, LOLSGD obtains much higher unseen accuracy than naive fine-tuning.

Table 16: Results of CoCoOp with ViT-B/32 on CI-FAR100. 50 classes are missing during training (50 seen classes). We treat the trained CoCoOp as a source model and apply our approaches.

| Methods | Overall | Unseen |
|---|---|---|
| Source (CLIP) | 64.18 | 62.42 |
| Naive Target | 65.94 | 42.66 |
| LOLSGD❄ | 69.59 | 51.78 |
| LOLSGD $+\mathcal{L}_{\text{distill}}$ ❄ | 67.13 | 49.08 |
| LOLSGD $+\mathcal{L}_{\text{rank}}$ ❄ | 69.99 | 52.54 |
| LOLSGD $+\mathcal{L}_{\text{rank}}$ +SE❄ | 72.31 | 58.38 |
| CoCoOp | 66.16 | 63.08 |
| CoCoOp + LOLSGD❄ | 72.58 | 64.20 |
| CoCoOp + LOLSGD $+\mathcal{L}_{\text{distill}}$ ❄ | 69.34 | 61.90 |
| CoCoOp + LOLSGD $+\mathcal{L}_{\text{rank}}$ ❄ | 72.77 | 64.36 |
| CoCoOp + LOLSGD $+\mathcal{L}_{\text{rank}}$ +SE❄ | 72.34 | 65.52 |
| Oracle | 83.10 | 82.62 |

Table 17: iWildCAM mean accuracy of 21 new locations. We extend cf. Table 6 to compare to the baseline "BN (stats) only", which update the means and variances in all BN layers.

| Methods | Overall | Seen | Unseen |
|---|---|---|---|
| Source | 35.71 | 35.74 | 26.43 |
| Naive Target | 35.38 | 51.08 | 1.90 |
| BN (stats) only | 30.18 | 29.69 | 23.25 |
| SGD❄ | 36.53 | 49.00 | 7.91 |
| SGD $+ \mathcal{L}_{\text{rank}}$ ❄ | 36.28 | 40.98 | 19.76 |
| SGD $+ \mathcal{L}_{\text{distill}}$ ❄ | **41.59** | 50.40 | 17.90 |
| LOLSGD❄ | 38.12 | 48.91 | 13.30 |
| LOLSGD $+\mathcal{L}_{\text{rank}}$ ❄ | 33.94 | 37.62 | 19.42 |
| LOLSGD $+\mathcal{L}_{\text{distill}}$ ❄ | 40.49 | 47.30 | **25.16** |
| LOLSGD $+\mathcal{L}_{\text{distill}}$ +SE❄ | 39.96 | 44.79 | **25.66** |

Table 18: Results on Office-Home with 30 seen and 35 unseen classes. We combine updating BN with partially tuning the last block of ResNet-50. The accuracy is averaged over 6 source-target pairs used in our paper.

| Methods / Acc. | Overall | Unseen |
|---|---|---|
| Source | 63.90 | 65.65 |
| Naive Target | 53.50 | 22.36 |
| BN (stats) only | 64.27 | 67.51 |
| BN (stats) only + Partially tune | 57.96 | 30.63 |
| BN (stats) | 57.60 | 32.36 |
| BN (stats) + Partially tune | 57.68 | 29.39 |
| Oracle | 72.01 | 66.54 |

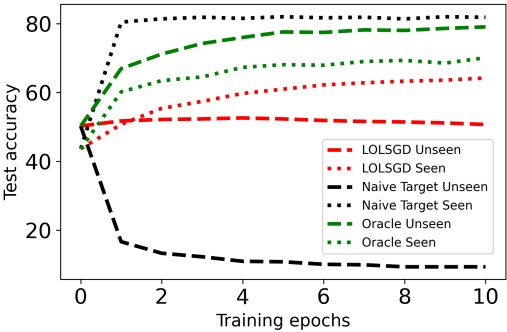

Table 19: An HT example showing that LOLSGD can cancel out undesired class biases learned by Naive Target. We use Ar → Cl domains on Office-Home

# E  Additional Discussions

## E.1  Limitations

In this paper, we introduce a novel and practical transfer learning problem, holistic transfer, that emphasizes the generalization to domain shifts for classes unseen in the target domain but seen in the source domain. We establish strong baselines and demonstrate the potential for simultaneously improving both seen and unseen target classes. One potential limitation is that we mainly focus on vision classification tasks. We leave the studies to image segmentation/object detection and natural language processing tasks as our future work. We also plan to explore better approaches for the disentanglement of domain styles and classes and to integrate our approach with other learning paradigms, like test-time adaptation.

## E.2  Potential Negative Societal Impact

The goal of our work is to introduce and study a practical transfer learning problem, holistic transfer. We provide strong baselines and analyze the problem on publicly available datasets, which are adjusted and split to meet our problem setting. As far as we know, our work does not introduce additional negative societal impacts compared to the standard transfer learning topics, like domain adaptation and out-of-distribution generalization.

### E.3 Computation Resources

We conduct our experiments on PyTorch and on NVIDIA V100 GPUs. On the Office-Home dataset, fine-tuning for 1 target domain with all the compared methods and random seeds takes roughly 36 hours on 1 GPU. Similar time consumption also applies to iWildCam and VTAB datasets. On the smaller FEMNIST dataset, it takes roughly 0.5 hours on 1 GPU to get the required results for 1 target domain. The whole experiment on iNaturalist Fungi takes roughly 0.5 on 1 GPU. In total, our experiments take roughly 1.3K GPU hours.

