# OpenReview forum: "Holistic Transfer: Towards Non-Disruptive Fine-Tuning with Partial Target Data"
_NeurIPS.cc/2023/Conference — NeurIPS 2023 poster_

### Official Review · Reviewer_HMzM · 2023-06-30

**Soundness:** 2 fair
**Presentation:** 2 fair
**Contribution:** 2 fair
**Rating:** 3
**Confidence:** 4

**Summary:**

In this paper, the authors tackled "holistic transfer" task in which imcomlete target data that only cover a part of class labels are given to fine-tune a pre-trained model. To gain generalization ability of the model to unseen-class data in the target domain, the proposed method adopts several techniques in the fine-tuning with the target data: leave-out local SGD, freezing a linear classifier, selective distillation, and feature rank regularization. The experimental results with several benchmark datasets demonstrate the effectiveness of the proposed method.

**Strengths:**

- The problem setting called holistic transfer is interesting and should be important when we consider practical ML applications.
- The proposed method works well in terms of boosting overall accuracy across several datasets in the experiments.

**Weaknesses:**

- This paper fairly lacks discussion on the comparison of the proposed task/method with related work.
    - The problem setting of holistic transfer is quite similar with partially zero-shot domain adaptation (PZDA) [R1]. Specifically, it can be seen as a supervised but source-free version of PZDA. Since a generative approach to source-free domain adaptation is somewhat common in the recent literature, using PZDA method with source-data generation seems to be a simple solution to holistic transfer.
        - [R1] "Partially Zero-shot Domain Adaptation from Incomplete Target Data with Missing Classes," WACV 2020.
    - Additionally, it should be also straightforward to use test-time adaptation or online domain adaptation methods to solve the holistic transfer task.
    - The performance of existing methods such as those raised above is not examined in the experiments, which makes the significance of the proposed method unclear. Comapring with domain generalization methods should be unfair, because they are essentially unable to utilize the information of the target domain.
- In the experiments, the performance of the proposed method in unseen classes is often par or even worse than that of the source model, which raises questions about the effectiveness of the proposed method for generalizability to unseen classes.
- The manuscript lacks several important topics such as related work and limitations, while the discription about the problem setting (section 2) is redundant.

**Questions:**

- Why did the authors not examine the existing methods such as test-time adaptation, online domain adaptation, or PZDA in the experiments?
- Can we say that the model is also successfully adapted to unseen classes even if the accuracy in those classes does not change from the source model? If yes, why?

**Limitations:**

- LOLSGD essentially requires that the number of classes is large enough to provide a sufficient variety of label subsets.

---

> ### Author Rebuttal · Authors · 2023-08-08
>
> **The manuscript lacks several important topics such as related work and limitations.** Due to the page limit, we leave the related work and limitations in the Supplementary (as mentioned in L343). We apologize if it was unclear, and we will clarify it in the final version.
>
> **Lacks discussion and comparison with PZDA, online domain adaptation, and test-time adaptation.**
> Thank you for the valuable comments. Our holistic transfer (HT) setting shares conceptual similarities with several machine learning paradigms (such as continual learning and domain adaptation, as discussed in Sect. 2, Table 2, and related work). We tried to include comprehensive discussions, but it appears we missed some. We appreciate your references to partially zero-shot domain adaptation (PZDA), test-time adaptation, and online adaptation, and we will surely cite and discuss them in the final version.
>
> The key difference between our HT and PZDA is that we consider a source-free setting, which is more practical, especially for models trained on large-scale datasets or scenarios with privacy concerns (L83-85). If the source data set is available during adaptation, the problem can indeed become much easier. We have discussed a solution at L148-153, which is to translate the source data into the target style.
>
> Our HT setting is also quite different from test-time adaptation and online adaptation. While both of them allow the source model to be dynamically adapted once new target data is collected, they are not meant to address our problem of adapting the source model’s holistic capability of recognizing $N$ classes to the target domain when the available target data only covers $N’<N$ classes. More specifically, test-time adaptation and online adaptation cannot directly tackle the situation where some of the target test classes are missing in the target training set (cf. Table 2). While both of them can *wait* till the model eventually see the missing target classes to adapt the model’s capability of those classes, the model may have already suffered a serious forgetting problem (please see our next response). In contrast, our HT setting aims to *proactively* adapt the model, even for some target test classes that are missing in the target training set.
>
> We think the source-data generation idea you suggested is very interesting and itself deserves a future paper since we believe source-data generation remains a challenging and ongoing research problem. We would like to emphasize that our main contributions are proposing the HT problem and establishing some most relevant baselines (L35, L346). We certainly do not claim that we have solved the problem perfectly. That said, we have experimented with source-data generation for adaptation and found that the gap between using real source images and generated ones is still large. We consider the OfficeHome dataset and apply the popular DeepInversion [C] on the two source models trained on Ar and Rw domains, respectively. We conduct a simple baseline for our HT problem by combining the real/generated source data with the partial target data for fine-tuning. As shown in Table R5 (the PDF in the global response), the unseen accuracy gap between using real/generated source data remains significant. Examples of the generated source images can be seen in Figure R2 (the PDF in the global response).
>
> [C] Dreaming to Distill: Data-free Knowledge Transfer via DeepInversion, CVPR 2020.
>
> **Comparing with domain generalization methods should be unfair, because they are essentially unable to utilize the information of the target domain.** We apologize for the confusion. While SWA and SWAD (L262-263) are initially proposed to improve model generalization, we use them to fine-tune the source model with the target data. We include them mainly because they share a conceptual similarity to our LOLSGD, in which they also take average over several models in training. We will clarify this.
>
> **... the performance of the proposed method in unseen classes is often par or even worse than that of the source model … Can we say that the model is also successfully adapted to unseen classes even if the accuracy in those classes does not change from the source model? If yes, why?** We appreciate your question. The ultimate goal of HT (as defined in the objective in Equation 1) is to make the adapted model’s performance close to the oracle model that is trained/fine-tuned on the full target data without missing classes. To claim we successfully resolve HT thus requires improving the accuracy for both the seen and unseen classes in the target domain. However, one should not treat maintaining the unseen class accuracy the same as the source model as a trivial task, as fine-tuning with only the seen class data degrades the unseen class accuracy drastically (Sect. 2.3). Therefore, even keeping the unseen accuracy while improving the seen accuracy is challenging, especially in the source-free setting. In our humble opinion, our methods have largely addressed this challenge (from a drastically degraded unseen accuracy to an unseen accuracy similar to the source model) and, in some cases, improved the unseen accuracy against the source model. After all, our main contributions are proposing the HT problem and investigating some most relevant baselines (L35, L346). We certainly do not claim that we have solved the problem perfectly, and we hope our contributions will establish the foundation for future research in HT.

---

> > ### Comment · Reviewer_HMzM · 2023-08-11
> > **Reply**
> >
> > Thanks for the authors' response. I have read it as well as all other reviews.
> >
> > I do not have any further question from my side, but I am still not fairly convinced on the comparison. (a similar concern seems to be also raised by QPVq and CeTn)
> > Specifically, since source-free DA and test-time adaptation are quite popular in the literature, we would naturally expect that they are examined in the experiments for comparison. However, the authors did not directly compare them with the proposed method, but instead used their own baselines, which are somewhat similar to them, as the authors stated "inspired by" or "similar to" in the response to QPVq. I think this is why several reviewers, including myself, commonly raise this concern.

---

> > > ### Author Response · Authors · 2023-08-19
> > > **Thank you for your repsponse. Further response.**
> > >
> > > Dear reviewer,
> > >
> > > Thank you for reading our rebuttal and other reviewers' comments. We are glad that you do not have further questions. We respond to your remaining concern as follows.
> > >
> > > We acknowledge that source-free domain adaptation (SFDA) and test-time adaptation (TTA) share some similarities to our holistic transfer (HT). For example, we also consider a source-free setting. However, we want to emphasize that **HT is addressing quite different challenges from theirs.** For instance, based on our investigation, the core challenges in HT are the **forgetting of unseen classes and the bias to seen classes in the target domain** (L103-110, Sect. 2.3). These challenges, in our humble opinion, drastically differ from the challenges and technical focus in SFDA and TTA, which are to recover the true labels for the unlabeled target data.
> > >
> > > Therefore, in considering what baselines to compare to, we focus on those which can potentially address the core challenges in HT. In our humble opinion, the baselines that we design in Sect. 3.2 and 3.3 address the forgetting problem in HT more closely than directly applying methods in SFDA and TTA whose goal is not to handle forgetting.
> > >
> > > We hope the above clarification addresses your concern *"However, the authors did not directly compare them with the proposed method, but instead used their own baselines, which are somewhat similar to them, as the authors stated "inspired by" or "similar to" in the response to QPVq."* In short, many of the techniques proposed in these related paradigms do not aim to address the core challenges in HT. Directly applying them is thus unlikely effective for HT and may be considered as an unfair/misleading comparison. Nevertheless, we do draw insights from them (e.g., some components in their techniques) to design our baselines.
> > >
> > > That said, in our final version, we will be happy to apply SFDA and TTA methods to HT while knowing that they may not address the challenges in HT.
> > >
> > > Best,
> > >
> > > Authors

---

### Official Review · Reviewer_CeTn · 2023-07-06

**Soundness:** 3 good
**Presentation:** 3 good
**Contribution:** 3 good
**Rating:** 7
**Confidence:** 3

**Summary:**

- This paper introduces a new setting: partial target data. A model is pretrained on a source domain with a set of classes. The model then has access to labels from a target domain, but only a subset of the classes. The goal is to do well on all classes (including the remaining, unseen classes) on the target domain
- This paper repurposes datasets such as Office-Home, iWildCam, VTAB, iNaturalist (fungi), and FEMNIST, for this task.
- They find that naive fine-tuning on the target does not do well. It does well on the seen classes, but poorly on unseen target classes
- They consider a wide range of methods. For example, tuning only the batch norm parameters
- They propose a method called LOLSGD, where they leave out a class and take gradient steps, and average this over the left-out class.
- They combine these with ideas such as selective distillation, feature rank regularization
- They show that their method does well on a wide range of datasets


**Strengths:**

- The problem seems very relevant, practical, and interesting. I’m not personally aware of other work in this space, but I might be missing parts of the related literature
- I think their datasets generally seem to make sense, and could be useful for people working on this problem
- Their explanation of why fine-tuning doesn't work very well makes sense
- They try a lot of potential improvements, including some of their own, which are novel. These methods seem to work well
- I like the selective distillation idea - one reason a model might do poorly in the partial target data setting, is that it might have much higher confidence on seen objects than unseen objects. Selective distillation might mitigate this issue, because the logits on unseen classes should be similar
- The experiments generally look interesting and sound.

My overall verdict is an accept. While there are some weaknesses, I think the paper would be a solid addition to the conference.


**Weaknesses:**

- Batchnorm seems to do well, better than I expected from the text in line 174. BN (stats only) does the best on unseen data in Table 3.  Consider running these on the other datasets too? Maybe a slightly more sophisticated version might work better.
- I don't understand the intuition for LOLSGD. It seems handwavy. I think it would be good to add a toy example where this method works. Or show a simple example where the method works. It's unclear why the gradients biased to certain classes will cancel out
- Their solutions seem nice and plausible, but not especially convincing. This is fine, since they're introducing a problem, and try out some reasonable methods. Future work can focus on better or more principled solutions.
- nit: I’m not a big fan of the name holistic transfer. There are many different transfer settings. What makes this more holistic than other types, for example OOD generalization? I think “partial target data setting”, or some other more specific naming choice would be better.
- There should be a more comprehensive related work, and it should be in the main paper. Can you compare to universal domain adaptation, open set domain adaptation, etc? I wonder if there is some setting that is similar to yours, even if I'm not aware of the literature, which is why my confidence is low. Please select the closest related settings and compare with them.


**Questions:**

- For feature regularization, can you add some details for exactly what matrix you’re computing the singular values (is it the matrix C?). Also, if you regularize ||diag(C^T C)||_2^2, then doesn’t that push down the singular values? Are you subtracting this term instead of adding it? Sorry for I’m missing something basic.
- For CLIP, do you initialize the head with the zero-shot initialized classifier?
- I wonder if tuning just the bottom couple of layers would work well. Lee et al., 2023, found that for shifts in x it can often be better to tune the bottom few layers. This might have less overfitting to the seen classes as well. Maybe it can be combined with batchnorm fine-tuning.
- LP-FT: I think it’s worth adding that it’s not designed for this scenario (it’s designed for the case where the classes from source / pretraining to target / fine-tuning are different, so you don’t have a “head”. Otherwise it looks like a bit of a strawman.
- nit: Line 174 says “Unfortunately, we found updating normalization layers is yet satisfying” - do you mean it’s not yet satisfying? That is, updating normalization layers doesn’t quite solve the partial target data setting?


**Limitations:**

No concerns

---

> ### Author Rebuttal · Authors · 2023-08-09
>
> Thank you for the positive and constructive feedback.
>
> **Batchnorm seems to do well.** We note that our ultimate goal is to achieve high overall accuracy, not merely unseen accuracy. Although BN (stats only) *maintains* the unseen accuracy well on OfficeHome, it cannot effectively improve the seen accuracy, resulting in worse overall accuracy (cf. Table 3). Following your suggestion, we apply BN (stats only) to iWildCAM (cf. Table 6). As in Table R2 (in the PDF of the global response), BN (stats only) maintains the unseen accuracy well but cannot improve the seen accuracy. That said, we agree that a more sophisticated, dedicatedly designed version of BN might perform better, and we leave it as future work.
>
> **More details and demonstrations of LOLSGD.** Thank you for the comment. Please be referred to the global response.
>
> **The naming of holistic transfer.** We apologize for the confusion. The word “holistic” closely reflects our setting — adapt a pre-trained classifier’s *holistic* capability of recognizing $N$ classes to the target domain, even when the available target data does not cover all the $N$ classes. (Please be referred to the first and third responses to 9DJT for details.) We will clarify this in the final version.
>
> We indeed have considered naming our setting with “partial target data,” but it may confuse our setting with a related problem named partial domain adaptation [A]. Partial domain adaptation also considers the situation where the target data set contains partial (i.e., $N'<N$) classes. However, after adaptation, it only aims to perform well on those $N'$ classes. In sharp contrast, our setting aims to perform well on all the $N$ classes after adaptation. We thus purposefully refrain from using the term "partial" to avoid confusion. That said, we remain open to discussion about the name and are willing to consider any adjustments based on your feedback.
>
> [A] Partial adversarial domain adaptation. ECCV 2018.
>
> **More related work.** Thank you for the comment. Our related work is in the Supplementary due to the page limit. We will strengthen it and include the most relevant part in the main paper.
>
> Our setting is highly related to domain adaptation (DA), whose objective is to adapt a model trained on a source domain (e.g., an image style) to perform well on a target domain (e.g., a different style). Depending on whether *labeled* target data is provided for adaptation, DA can be categorized into supervised, semi-supervised, and unsupervised settings, and we consider the supervised setting (L81).
>
> Another way to categorize DA works is by the relationship between the label spaces of the source and target data. The standard closed-set setting assumes that the source and target data cover the same $N$ classes. That is, when one prepares the target data set for adaptation, all classes are expected to be present. However, in practice, this is not trivial and often infeasible for an end-user: collecting data can be quite laborious, especially when $N$ is huge. (Please be referred to the first response to 9DJT for details.) The proposed setting aims to release this constraint, making the data preparation for adaptation *much simpler* for end-users.
>
> At first glance, our setting seems quite similar to partial DA [A], which also considers the situation
> where the target data set contains part (i.e., $N’<N$) of the source classes. However, the goal of partial DA is to perform well only on those $N'$ classes after adaptation. In contrast, our setting aims to adapt the source model’s holistic capability of recognizing $N$ classes to the target domain. This is why in Table 1 we explicitly separate the target data set into one for training ($N’<N$ classes) and one for testing ($N$ classes). Our study in Sect. 2.3 showcases the challenges of this new setting: standard fine-tuning would simply degrade the model’s capability on the $N - N’$ unseen target classes.
>
> Open-set and universal DA consider a different scenario where the target data set contains additional “unknown” classes that do not appear in the source data. The goal is thus to equip the adapted model with the ability to predict “unknown” for those data. We argue that this setting is orthogonal but can be compatible with ours, and we leave the combination as future work.
>
> Please also see the first response to HMzM.
>
> **Feature regularization.** Yes, Figure 4 shows the singular values of matrix C, which is the covariance matrix of feature vectors. Ours has more *uniform* singular values. That said, in training, we do not explicitly compute the singular values and regularize them. The regularization term $||diag(C^\top C)||_2^2$ we use is inspired by [15] and [B], which is *added* to the loss function. While looking a bit counterintuitive, [B] proved that this regularizer has the effect of penalizing the variance among the singular values, thus discouraging the tail singular values from collapsing to 0, mitigating dimensional collapse. We apologize that we missed this detail in the manuscript, and we will clarify it in the final version.
>
> [B] Towards Understanding and Mitigating Dimensional Collapse in Heterogeneous Federated Learning, ICLR 2023.
>
> **CLIP initialization.** Yes, we initialize the classification head with class names’ embeddings extracted from CLIP’s text encoder. After that, we drop the text encoder.
>
> **Tuning the bottom couple of layers.** We experiment with fine-tuning the BN layers and the first (or last) block of the source ResNet-50 on OfficeHome. As shown in Table R4, fine-tuning parts of the layers drastically degrades the unseen accuracy. This showcases the challenge of holistic transfer. That is, even with just parts of the model being fine-tuned, the model can easily fit the seen classes and suffers from forgetting the unseen classes.
>
> **LP-FT.** We will do so.
>
> **Line 174.** You are right. We meant updating normalization layers does not solve holistic transfer. We will correct the typo.

---

> ### Comment · Reviewer_CeTn · 2023-08-15
> **Thanks for the response**
>
> I have read all the other reviews and the author response. I think the paper introduces an interesting problem, and has some nice analysis. So I think it is above the NeurIPS bar. But I could understand if the paper doesn't get in this time.
>
> The other reviewers mentioned that it would be nice to add cases where the target dataset contains classes that aren't present in the source dataset. I don't think this is necessary. I actually prefer the author's setting because it cleanly examines an important problem - if we have some knowledge about classes 1 to N, but then fine-tune on some data with missing classes, how do we preserve our knowledge of the missing classes? Adding unseen classes into the mix complicates the problem.
>
> That said, I'm still not a fan of the name "holistic" transfer, and I suspect that may be why the reviews are asking for this change. After reading the author response, it is still not clear to me why this is more "holistic" than other paradigms. Maybe something along the lines of "class extrapolation"? This might capture the fact that the original model may be CLIP (self-supervised) or pretrained on a source distribution - it doesn't matter. The point is it has some inductive bias for classes 1 to N, is prompted using a subset of classes, and should extrapolate to new classes.
>
> The rebuttal answers most of my questions. Thank you for explaining the connection with various related areas of research. Unfortunately, I haven't had the time to re-look into the detailed intuitions of LOLSGD too closely. I'd suggest adding a toy dataset or example where this method works (e.g., with mixtures of Gaussians, etc).

---

> > ### Author Response · Authors · 2023-08-19
> > **Thank you for your repsponse**
> >
> > Dear reviewer,
> >
> > Thank you for your valuable response. We are glad that our rebuttal addressed most of your questions. We are pleased you recognize our interesting problem setup, nice analysis, and our paper above the acceptance bar. We also appreciate your support in our current setting.
> >
> > Regarding the name "holistic" transfer, thank you for clarifying your concern and providing further suggestions. We apologize for the confusion --- **We certainly do not claim that our problem setting is more "holistic" than other transfer learning and domain adaptation paradigms, and we will clarify it.** We use "holistic" mainly to emphasize that we intend to transfer the *source classifier's "holistic" capability* --- i.e., being able to recognize $N$ classes --- even when the available target data set has missing classes.
> >
> > To address the confusion. We will consider changing the naming. For instance, "Towards Holistic Transfer of Classifier Capabilities: Non-Disruptive Fine-Tuning with Partial Target Data." We appreciate your suggestion (i.e., class extrapolation) and will certainly consider it.
> >
> > For LOLSGD, please kindly be referred to our global response when you have time. We will be happy to answer your further questions if there are any. We will incorporate the detailed intuition and evidence in our global response into our final version. We also appreciate that you gave us more details about the toy example, and we will add it to our final version.
> >
> > Best,
> >
> > Authors

---

### Official Review · Reviewer_9DJT · 2023-07-06

**Soundness:** 2 fair
**Presentation:** 2 fair
**Contribution:** 2 fair
**Rating:** 5
**Confidence:** 4

**Summary:**

This paper studies a learning problem, called Holistic Transfer, which involves the adaptation of a pre-trained source model capable of classifying a wide range of objects, to a target domain using data that covers only a partial label space. To solve this problem,

**Strengths:**

1. Distribution shift exists everywhere in real applications.
2. Experiments show the effectiveness of the proposed method.

**Weaknesses:**

1. This paper is not well-structured. Considering that the learning setup is proposed in this paper. It is more important to convince me the new setup is valuable. Now the introduction is too short and evidence is not convincing.
2. The learning setup is somewhat trivial. Researchers initially noticed distribution shift problem because the distribution shift is from sampling bias. It is okay to assume that the distribution of labels is different. In this case, sampling bias exists. It is confusing to assume that the marginal distribution of $T$ and $T^*$ is the same one.
3.  The name "Holistic Transfer" is also confusing for me. What is "Holistic" means?

**Questions:**

1. This paper is not well-structured. Considering that the learning setup is proposed in this paper. It is more important to convince me the new setup is valuable. Now the introduction is too short and evidence is not convincing.
2. The learning setup is somewhat trivial. Researchers initially noticed distribution shift problem because the distribution shift is from sampling bias. It is okay to assume that the distribution of labels is different. In this case, sampling bias exists. It is confusing to assume that the marginal distribution of $T$ and $T^*$ is the same one.
3.  The name "Holistic Transfer" is also confusing for me. What is "Holistic" means?

**Limitations:**

Yes

---

> ### Author Rebuttal · Authors · 2023-08-09
>
> **It is more important to convince me the new setup is valuable. Now the introduction is too short and evidence is not convincing.** We apologize for not making the motivation clear. We provide a detailed motivation as follows. We will incorporate it in the final version to expand the introduction.
>
> The proposed setting considers the practical scenario where *an end-user wants to adapt a pre-trained model’s capability of recognizing $N$ classes to the target domain* (L16-19). In the literature, this typically requires preparing a target data set that covers all the $N$ classes, which can be challenging or even *unrealistic*, especially when $N$ is huge (L20-21). The proposed setting aims to release this constraint, making the data preparation for fine-tuning *much simpler* for end-users.
>
> More specifically, in most of the literature on domain adaptation, the target data set has been “pre-prepared” to cover all the $N$ classes of the pre-trained source model. However, in practice, this is not trivial and often infeasible for an end-user: collecting data can be quite laborious and costly. Take wildlife monitoring as an example. Data are often collected passively (e.g., via smart camera traps), waiting for animals to appear. As a result, when a smart camera trap is redeployed to a new location and requires adaptation, it is hard to prepare a complete target dataset that contains all the animal species of interest. This raises a dilemma: *should one wait for the data to be fully collected, even if it means sacrificing the model's performance in the meantime, or update the model right away with incomplete data, accepting the risk of compromising certain capabilities?* Our paper aims to address this dilemma by delving into an unexplored  Holistic Transfer (HT) setting: adapting a pre-trained classifier's *holistic* capability of recognizing $N$ classes to the target domain, using target data that covers only a subset of $N'<N$ classes.
>
> It is worth noting that HT fundamentally differs from partial domain adaptation [A]. While partial domain adaptation also considers the situation where the target data set only contains $N'<N$ classes, its goal is only to perform well on those $N'$ classes after adaptation. In sharp contrast, HT aims to perform well on all the $N$ classes after adaptation. This is why our benchmarks in Table 1 explicitly separate the target data set into the one used for training ($N’<N$ classes) and the one used for testing ($N$ classes). Our study in Sect. 2.3 showcases the challenges of this new setting: standard fine-tuning would simply degrade the model’s capability on the $N - N’$ unseen target classes.
>
> We hope the above paragraphs address your concern, and we will be happy to provide more information in the discussion period.
>
> [A] Cao, Zhangjie, et al. "Partial adversarial domain adaptation." Proceedings of the European conference on computer vision (ECCV). 2018.
>
> **The learning setup is somewhat trivial … It is confusing to assume that the marginal distribution is the same one.** Sorry for the confusion. About the relationships between P_T and P_T* at L60, we should have written P_T(x|y) = P_T*(x|y), not P_T(x) = P_T*(x). That is, we assume that the data distributions in T and T* are the same per class, not marginally. We will correct this accordingly.
>
> We hope this addresses your concern that *the learning setup is somewhat trivial.* If not, we kindly ask for more details about your comment, and we will be happy to address it in the discussion period.
>
> **The name "Holistic Transfer" is also confusing for me.** Sorry for the confusion. Following the motivations we provide above, the goal of our new setting is to adapt a pre-trained classifier’s *holistic* capability of recognizing $N$ classes to the target domain, even when the available target data does not cover all the $N$ classes. This is why we call our setting “holistic transfer.”
> We will clarify this in the final version.
>
> That said,  we remain open to discussion with you regarding the name and are willing to consider any adjustments based on your feedback.
>
> *In light of our clarifications, we would like to ask if you are willing to reconsider your score and if there are any new concerns or additional questions we can respond to!*

---

> > ### Comment · Reviewer_9DJT · 2023-08-19
> > **Response to authors**
> >
> > Thanks for the detailed reply. My concerns have been partially addressed. I also read the others' reviews, and find Reviewer QPVq also find the setup in this paper not very exciting. I think this paper can raise discussions and is somewhat valuable to the community. However, the writing qualities need to be improved and now the version is not ready for publication. I am inclined to raise the score to 5, and I hope that the new version could make several improvements in writing quality (especially learning setup) if the paper can be accepted.

---

> > > ### Author Response · Authors · 2023-08-19
> > > **Thank you for your repsponse**
> > >
> > > Dear reviewer,
> > >
> > > Thank you for your time reading our rebuttal. We are glad that our rebuttal has addressed some of your concerns. **We are happy to know that you are willing to raise your score to 5.**
> > >
> > > Regarding Reviewer QPVq's concern about the paper setup, we have addressed it in the corresponding response. Regarding your concern about the writing quality, specifically the too-short introduction and learning setup, we will certainly improve it, including incorporating our rebuttal properly.
> > >
> > > Please do not hesitate to let us know if there are any new concerns or additional questions we can respond to!
> > >
> > > Best,
> > >
> > > Authors

---

> > > > ### Author Response · Authors · 2023-08-21
> > > > **Thank you again**
> > > >
> > > > Dear reviewer,
> > > >
> > > > Once again, we are glad to know that our rebuttal addressed some of your concerns. We appreciate your willingness to increase the rating **from 4 to 5.** While it has not been shown, we look forward to it. Thank you.
> > > >
> > > > Best,
> > > > Authors

---

> > > > > ### Comment · Reviewer_9DJT · 2023-08-22
> > > > > **Response to authors**
> > > > >
> > > > > I appreciate the authors' efforts in the reponse, and now the ratings has been changed to 5.

---

### Official Review · Reviewer_tg8m · 2023-07-06

**Soundness:** 3 good
**Presentation:** 3 good
**Contribution:** 3 good
**Rating:** 7
**Confidence:** 4

**Summary:**

This paper proposes "Holistic Transfer" as an important problem and also provides some solutions to it.  "Holistic Transfer" handles the situation when adapting a pre-trained source model (e.g. with 1000 classes) to a target domain (e.g. with 100 classes), but there are only data for part of the target domain (e.g. only with target domain data for 50 classes and no target domain data for the other 50 classes). Taking classification as an example, it assumes that all the target domain classes are in the source data, but may not be all in the target domain data.  Usual fine-tuning can improve the performance on seen classes (present in the target domain data), but may destroy the performance on unseen classes (not present in the target domain data).

The paper proposes some solutions in handling these situations, aiming to preserve the performance on unseen classes. In the proposed approach, the changes in the target domain are decomposed into two types of changes: "style" change and class change.  Style change can be learned via the seen classes of the target domain, assuming that the unsee classes are with the same style change. The information from the unseen classes are preserved via distillation and feature rank regularization.

Experiments were conducted on various datasets to validate the approach.

**Strengths:**

The problem proposed in the paper is valid and practically useful problem. As large foundation models gains popularity, how to adapt those models to specialized target domains effectively is of practical importance.  The paper handles the situation when the data for some classes of the target domain classes are missing.

The proposed approach seems to be sound. It decomposes the changes in the target domain into two types of changes: "style" change and class change.  Style change can be learned via the seen classes of the target domain, assuming that the unsee classes are with the same style change. The information from the unseen classes are preserved via distillation and feature rank regularization.

The effectiveness of the proposed approach is validated on various datasets.

**Weaknesses:**

The paper assumes that both the "seen" and "unseen" classes from the target domain are subsets of the classes in the source domain.  It may happen that some "seen" classes in the target domain are not in the classes of the source domain.  It would interesting to see how these cases are handled.

**Questions:**

One question regarding the results in Table 1, from some rows, e.g. the row of "BN (stats) only", the "Unseen" performance is better than "Overall" performance, why is that?

**Limitations:**

yes

---

> ### Author Rebuttal · Authors · 2023-08-09
>
> Thank you for the positive and constructive feedback on our paper.
>
> **The paper assumes that both the "seen" and "unseen" classes from the target domain are subsets of the classes in the source domain. It may happen that some "seen" classes in the target domain are not in the classes of the source domain. It would interesting to see how these cases are handled.**
>
> Thank you for the comment. If some “seen” classes in the target domain are not in the source domain, this will require expanding the label space of the model. A simple baseline would be first training the classification weights for those “seen” classes from scratch while keeping all other model components intact. Then, we can apply our holistic transfer approach to the expanded model. A more sophisticated solution would involve techniques from continual learning, or more specifically, class-incremental learning. This machine learning paradigm aims to expand the label space of a model. We leave a suitable combination of our approach and techniques from class-incremental learning as future work.
>
> **One question regarding the results in Table 1, from some rows, e.g. the row of "BN (stats) only", the "Unseen" performance is better than "Overall" performance, why is that?**
>
> We surmise this is because the classes are not equally difficult for classification. In some cases, before we perform adaptation, we can already see that the unseen class accuracy of the source model is higher than its overall accuracy, meaning that those unseen classes are inherently easier than the seen classes. Some methods (e.g., BN (stats) only) can better keep the unseen accuracy close to the source model with less forgetting. However, they cannot adapt the seen classes effectively to the target domain. Therefore, the results of these methods generally follow the trend in the source model with the unseen accuracy higher than the overall accuracy. In contrast, some methods (e.g., LP-FT) can better adapt the seen classes to the target domain but suffer from serious forgetting of the unseen classes. These methods thus have lower unseen accuracy.

---

### Official Review · Reviewer_QPVq · 2023-07-09

**Soundness:** 2 fair
**Presentation:** 3 good
**Contribution:** 2 fair
**Rating:** 3
**Confidence:** 4

**Summary:**

This paper proposes a problem of fine-tuning the source model with partial target data, where the source and target distributions have covariate shifts and the target test data contain classes unseen in the target training data (also called Holistic Transfer, HT, in this paper). It proposes Leave-Out Local SGD (LOLSGD) to disentangle domain gradients from classification gradients during training and several regularizations to preserve class relationships. In the experimental part, this paper builds some holistic transfer benchmarks based on existing domain adaptation and fine-tuning datasets and evaluates the proposed methods.

**Strengths:**

* This paper generally writes well. The analysis of the dilemma in the problem makes sense, and the proposed methods are easy to understand.

* This paper builds some benchmarks for the proposed holistic transfer problem and evaluates some methods on them. Some of the evaluations and results may help future works on related topics.


**Weaknesses:**

* The proposed problem does not seem very interesting or practical to me. In my opinion, a practical problem should be simple and realistic. For example, in the problem of fine-tuning, we only need to have a pre-trained model and some target training data. However, in the proposed holistic transfer problem, there exist too many constraints and assumptions (for example, it requires that the source domain covers all the labels in the target domain), which may not be satisfied in real applications. In Table 1, the authors give some examples of the HT problem, but they actually contain two different settings, making the definition of the problem a bit confusing. The first three examples are variants of source-free domain adaptation, with partial classes in target training data. The last two examples are fine-tuning of CLIP models, and CLIP models are actually self-supervised pre-trained, so it does not fit the definitions in Section 2.1. Besides, the problem of fine-tuning CLIP and testing on unseen classes has already been proposed by previous works [1].

* The technical novelty of the proposed methods is limited. The LOLSGD seems similar to local SGD and meta-learning. It is unsure why its design can disentangle covariate shifts. What is its difference from using a larger batch size for SGD? The class relationship regularizations and ensemble methods also seem to be existing techniques.

* This paper only discusses the problem itself and evaluates some proposed techniques. Some important related topics and methods are not discussed and compared in this paper. The source-free domain adaptation [2] and its following works are highly related but are not compared. Some fine-tuning methods may also solve the problem but are also missing, such as [3][4]. For the CLIP fine-tuning, some existing works already explore the problem of unseen classes [1], which are not mentioned.

[1] Conditional Prompt Learning for Vision-Language Models.

[2] Do We Really Need to Access the Source Data? Source Hypothesis Transfer for Unsupervised Domain Adaptation.

[3] DELTA: Deep Learning Transfer using Feature Map with Attention for Convolutional Networks.

[4] Catastrophic Forgetting Meets Negative Transfer: Batch Spectral Shrinkage for Safe Transfer Learning.



**Questions:**

Some questions are listed in ‘Weaknesses’, and some more questions here:

* How would the performance change if the number of classes in the target training data is changed?

* Would the increase in unseen classes sacrifice the performance on seen classes?

* In Equation (5), how can we get argmin_theta?

**Limitations:**

The authors point out the limitation that this paper mainly focuses on vision classification tasks and leave the studies to image segmentation/object detection and natural language processing tasks as their future work.

---

> ### Author Rebuttal · Authors · 2023-08-09
>
> **The proposed problem does not seem very interesting or practical … too many constraints and assumptions … Table 1 actually contains two different settings …** We want to reiterate that the proposed problem is fairly *practical* and *realistic*. It considers a *practical* scenario where *an end-user wants to adapt a pre-trained model’s capability of recognizing $N$ classes to the target domain.* In the literature, this typically requires preparing a target data set that covers all the $N$ classes, which can be challenging or even *unrealistic*, especially when $N$ is huge (L1-3, L16-21). The proposed problem releases this constraint, making the data preparation for fine-tuning *much simpler* for end-users.
>
> We apologize if Sect. 2.1 gives the wrong impression that our proposed problem introduces many constraints and assumptions, and we will carefully refine it. Sect. 2.1 is meant to describe our setting and ground it in the literature formally. The assumptions we mentioned (L54-63) either follow existing works or help clarify our setting. For instance, *the source domain covers all the labels in the target domain* is a direct property of the aforementioned practical scenario; the assumptions in L58-63 actually *relax* the assumption made in L124-125 by existing works. We should not have used the word “constraints” at L69. Indeed, L70-85 simply describes the properties and challenges of the proposed problem. We consider a more realistic source-free setting (L83-85) as it removes the need to access the source data.
>
> We also apologize for the confusion in Table 1. All five cases strictly follow the setting of holistic transfer — adapt a pre-trained model when the available target data only contain partial classes. We use CLIP mainly to extend our study to VTAB (Table 7) and iNaturalist (Table 9), for which we have no explicit source data to pre-train the model. We use CLIP’s zero-shot capability only to construct the source classifier, and we disregard its text encoder afterward. In other words, our problem definition has nothing to do with CLIP; our methods are not designed specifically for CLIP. We will clarify this in the final version.
>
> **Technical novelty.** We want to emphasize that our main contribution is studying the holistic transfer (HT) problem. The novelties are thus not merely technical but involve the investigation and understanding of the problem and the construction of benchmarks. For instance, Sect. 2.3 reveals the challenges of HT, which leads to the two directions of solutions (Sect. 3.2 and 3.3). Within each, instead of directly proposing new techniques, we deliberately seek existing techniques (initially developed for other problems) that can potentially address the challenges. While knowing that this approach might not result in many technical novelties, we believe it is crucial and valuable, as it helps establish the foundation of HT and connect it to the existing machine learning techniques.
>
> For the novelty and detail of our proposed LOLSGD, please be referred to the global response.
>
> **Related topics and methods.** Thank you for the comment and references. We will cite them and add more discussions on related topics. We did include the frozen linear classifier (L210-212) inspired by source-free adaptation (SFDA) in our experiments. Nevertheless, SFDA does not address adapting a pre-trained model using partially available target data. [3] and [4] aim to fine-tune pre-trained backbones for new downstream tasks. Their focus is to reduce over-fitting, for example, via distillation [3] or regularization [4]. Therefore, they are similar to the selective distillation and feature rank regularization (L213-226) in our experiments, which, however, could not fully solve the HT problem.
>
> For CLIP fine-tuning, we want to reiterate that CLIP is not our main focus. We use CLIP only to construct the source models for the VTAB and iNaturalist experiments. That said, we follow your comment to compare to CoCoOp [1] in these experiments. CoCoOp [1] fine-tunes CLIP by training a meta-net to condition the classification weights (i.e., the fully-connected layer) on the input images. In other words, CoCoOp freezes the visual features but changes the classifier weights by minimizing the standard cross-entropy loss. In contrast, our approach freezes the classifier weights but adapts the visual features by minimizing the loss designed specifically for HT.
>
> We conduct two experiments: 1) CoCoOp alone for the HT problem, and 2) combining CoCoOp with our approach. For 2), we take the resulting model after CoCoOp as the improved source model and further adapt the feature. We report the result on the CIFAR-100 task in VTAB (cf. Table 7). As shown in Table R1 (see the PDF in the global response), CoCoOp alone performs well on unseen classes, but it improves the overall accuracy only marginally. Combining both approaches, we can obtain better accuracy in unseen and seen classes, leading to the best overall accuracy.
>
> **Number of seen and unseen classes.** We conduct an experiment on OfficeHome (cf. Table 3). Starting from 10 seen classes, we gradually include more seen classes. As shown in Table R3, the overall accuracy consistently improves when the number of seen classes increases.
>
> To answer whether the increase of unseen classes (equivalently, the decrease of seen classes) would sacrifice the performance of seen classes, we calculate the seen class accuracy on the 10 commonly seen classes — this makes the accuracy comparable across settings. As shown in Table R3, the accuracy on the 10 commonly seen classes actually improves when we decrease the number of seen classes. We surmise that this is because when fewer seen classes are available, the adapted model will focus more on the 10 commonly seen classes.
>
> **Equation (5).** The argmin_theta means training the model on a locally sampled class subset in LOLSGD. We minimize the loss using SGD for several steps. We will clarify this.

---

> > ### Comment · Reviewer_QPVq · 2023-08-17
> > **Thanks for the response**
> >
> > I have read the comments from other reviewers and the author response, and some concerns have been addressed. The remaining concerns mainly lie in the problem setting and the evaluation.
> >
> > I am still not fully convinced of the novelty or practical value of the proposed holistic transfer problem. It is much less flexible than standard fine-tuning. Compared with source-free DA, fewer classes are needed, but the accuracy may also be sacrificed. It is unclear whether this intermediate state is valuable to explore. Furthermore, the setting is currently not unified for different pre-trained models in the experiments. (For the ImageNet pre-trained model, it is firstly trained on a small source domain and then transferred to the target domain, which is more like ‘partial source-free DA’. For the CLIP pre-trained model, the model is directly transferred to the target domain, which is more like ‘fine-tuning’ and has been explored before.)
> >
> > In evaluation, there are some other related topics mentioned by the reviewers, such as source-free DA, fine-tuning, and test-time adaptation, which can also potentially solve holistic transfer. But it seems that these topics or methods are not properly discussed or compared.

---

> > > ### Author Response · Authors · 2023-08-20
> > > **Thank you for your response. Further response (Part 1).**
> > >
> > > Dear reviewer,
> > >
> > > Thank you for reading our rebuttal and other reviewers' comments. We are glad that we have addressed some of your concerns. We respond to your remaining concerns as follows.
> > >
> > >  **I am still not fully convinced of the novelty or practical value of the proposed holistic transfer problem. It is much less flexible than standard fine-tuning.**
> > >
> > > We respectfully think comparing standard fine-tuning (a technique) and the proposed holistic transfer problem (a problem setting) may not make sense. In our humble opinion, the counterpart of our holistic transfer problem is the setting where the target data set covers all the classes of interest. In this counterpart setting, standard fine-tuning is certainly the go-to approach. However, in the proposed setting where the target data set has missing classes, standard fine-tuning will simply suffer forgetting, as evidenced in Sect. 2.3.
> > >
> > > Regarding which setting is more flexible, we respectfully disagree that our setting is less flexible than the counterpart setting. Similar to few-shot learning vs. many-shot learning or semi-supervised learning vs. fully supervised learning, we consider the scenario where the available data does not meet the requirement of the standard technique. Specifically, one may not obtain a comprehensive target data set covering all the classes of interest. For instance, in our response to Reviewer 9DJT, we gave one practical example ---
> > > camera traps --- where the data is collected passively, waiting for animals to appear. It is thus hard to compile a comprehensive target data set for adapting the model. Our setting enables adapting a pre-trained model to the target domain even under such a partial data situation, which we consider more flexible than the counterpart setting.
> > >
> > > **Compared with source-free DA, fewer classes are needed, but the accuracy may also be sacrificed. It is unclear whether this intermediate state is valuable to explore.**
> > >
> > > We acknowledge that if one can obtain a target data set covering all the classes of interest, the accuracy after adaptation should be higher. However, in some practical scenarios, collecting such a data set can be laborious, costly, and even infeasible, and our holistic transfer (HT) setting aims to tackle such a situation. In other words, we do not view HT merely as needing fewer classes but as a setting to address the problem where only partial class data are available.
> > >
> > > We respectfully think this is reminiscent of few-shot learning vs. many-shot learning or semi-supervised learning vs. fully supervised learning. It is well-known that many-shot learning and fully supervised learning can lead to better accuracy, but they require significant effort in data collection. Few-shot learning and semi-supervised learning aim to address the problem when such an effort cannot be met, and the community recognizes their values even if we know that their accuracy might be sacrificed.
> > >
> > > **Furthermore, the setting is currently not unified for different pre-trained models in the experiments.**
> > >
> > > We apologize if our rebuttal has not fully addressed this concern. We acknowledge that we applied different approaches to obtain the pre-trained models, and the main reason is that there is no clear source data set for VTAB and iNaturalist. Nevertheless, both 1) ImageNet pre-training followed by training on a source domain and 2) CLIP pre-training lead to a pre-trained classifier capable of classifying $N$ classes that cover the target label space. (We note that we drop the CLIP text encoder after we obtain the $N$ class names’ embedding.) The core problem of HT is then to adapt such a pre-trained classifier capable of recognizing $N$ classes to the target domain using target data that covers only partial (i.e., $N’ < N$) classes. In other words, regardless of how the pre-trained classifier was built, we study HT in a unified setting.
> > >
> > > **For the CLIP pre-trained model ... has been explored before.**
> > >
> > > We apologize if our rebuttal has not fully addressed this concern. We want to reiterate that CLIP fine-tuning is not our main focus. We use CLIP only to construct the pre-trained models for the VTAB and iNaturalist experiments. We drop the CLIP text encoder after we obtain the classification head with class names’ embeddings. Thus, our setting is not the same as CoCoOp [1]. That said, we will tone down the use of CLIP in our final version --- after all, CLIP is mainly used to obtain pre-trained models for some experiments, which can be replaced with other (future) foundation models.

---

> > > > ### Author Response · Authors · 2023-08-20
> > > > **Thank you for your response. Further response (Part 2).**
> > > >
> > > > **In evaluation, there are some other related topics mentioned by the reviewers, such as source-free DA, fine-tuning, and test-time adaptation, which can also potentially solve holistic transfer. But it seems that these topics or methods are not properly discussed or compared.**
> > > >
> > > > We acknowledge that source-free domain adaptation (SFDA) and test-time adaptation (TTA) share some similarities to our holistic transfer (HT). For example, we also consider a source-free setting. We have provided additional discussions on these topics in the rebuttal (please see our responses to Reviewer HMzM and CeTn), and we will certainly incorporate them into the final version.
> > > >
> > > > However, we want to emphasize that **HT is addressing quite different challenges from theirs.** For instance, based on our investigation, the core challenges in HT are the **forgetting of unseen classes and the bias to seen classes in the target domain** (L103-110, Sect. 2.3). These challenges, in our humble opinion, drastically differ from the challenges and technical focus in SFDA and TTA, which are to obtain high-quality pseudo-labels for the unlabeled target data. We thus respectfully do not think their techniques can solve holistic transfer.
> > > >
> > > > Therefore, in considering what baselines to compare to, we focus on those which can potentially address the core challenges in HT. In our humble opinion, the baselines that we design in Sect. 3.2 and 3.3 address the forgetting problem in HT more closely than directly applying methods in SFDA and TTA whose goal is not to handle forgetting.
> > > >
> > > > Regarding fine-tuning, we already discussed and compared it (sect. 2.3 and the “Naive Target” row in Tables 3, 5, 6, and 7). Regarding your
> > > > references [3] and [4], they aim to fine-tune pre-trained backbones for new downstream tasks with full-class data. Their focus is thus to reduce over-fitting rather than prevent forgetting unseen classes. (We have discussed this in the rebuttal.)
> > > >
> > > > That said, in our final version, we will be happy to apply SFDA, TTA, and more fine-tuning methods to HT while knowing that they may not address the challenges in HT. Overall, we humbly believe that we have compared to more direct baseline methods that can potentially address the HT problem in the manuscript.

---

### Author Rebuttal · Authors · 2023-08-09

We thank the reviewers for their valuable comments. We are glad that the reviewers found the proposed problem “interesting”, “important” (Reviewer HMzM), and “valid and practically useful” (Reviewer tg8m, CeTn); the proposed method “works well” (Reviewer HMzM, tg8m, 9DJT, CeTn, 9DJT); the experiments generally look “interesting and sound” (Reviewer CeTn). We address the comments about LOLSGD in this global response (raised by Reviewer QPVq and CeTn) and address other comments raised by individual reviewers separately. We also include a one-page PDF with requested experiments, tables, and figures. We will incorporate all the feedback in the final version.

**(Reviewer QPVq) The LOLSGD seems similar to local SGD and meta-learning. It is unsure why its design can disentangle covariate shifts. What is its difference from using a larger batch size for SGD?** Our LOLSGD is fundamentally different from meta-learning and SGD with a large batch size. In our holistic transfer (HT) problem (see Sect. 3), fine-tuning is affected by the covariate shift (from the source to the target) and the disruptive concept shift (from classifying all the classes to classifying the seen classes). Our LOLSGD aims to disentangle them or, more precisely, reduce the disruptive concept shift by subsampling multiple datasets that each contain a subset of the seen classes. When updating the model separately in parallel with these subsampled datasets, each updated model is affected by a different concept shift but a shared covariate shift. By averaging these models, LOLSGD can potentially cancel out the disruptive concept shifts and strengthen the shared covariate shift (L191-196). We provide more evidence that LOLSGD can cancel out the disruptive concept shifts in the following response to Reviewer CeTn.

We argue that this is fundamentally different from meta-learning, whose goal is to learn a meta-modal that can be easily applied to a future task (e.g., a few-shot classification task). While meta-learning also subsamples its meta-training set, it is mainly to simulate multiple future tasks for learning the meta-model, not to cancel out unwanted gradients. We also argue that LOLSGD is fundamentally different from SGD with a large batch size. Concretely, without the strategic subsampling in LOLSGD, SGD with a large batch size will not create multiple models from which we can potentially cancel out the disruptive concept shifts.

Our LOLSGD is inspired by local SGD, as mentioned in L177-179. Our key contribution is novel usage. Local SGD was initially proposed for distributed learning, in which the training data are decentralized by default. The goal is mainly for reducing the communication overhead of large-scale training. In contrast, in LOLSGD, the target training data set is not decentralized initially, but we strategically subsample it to simulate different concept shifts.

**(Reviewer CeTn) I don't understand the intuition for LOLSGD. I think it would be good to add a toy example where this method works. Or show a simple example where the method works. It's unclear why the gradients biased to certain classes will cancel out.** We apologize if we did not make the intuition clear. We have provided some more details in the above response to
Reviewer QPVq. Please kindly be referred to it.

To give more evidence for the canceling-out effect, we compare the seen class accuracy among 1) naive fine-tuning with the partial target data, 2) LOLSGD, and 3) fine-tuning with the full target data (i.e., the oracle at L148-153). As shown in Figure R1 (the PDF in the global response), the seen class accuracy of naive fine-tuning (black dotted line) exceeds the oracle (green dotted line), indicating that naive fine-tuning learns undesired concept shifts towards seen classes, leading to an unreasonable accuracy. In contrast, the seen accuracy of LOLSGD (red dotted line) consistently stays below the oracle, indicating that undesired concept shifts are reduced. As a result, LOLSGD obtains much higher unseen accuracy than naive fine-tuning.

---

### Decision · Program_Chairs · 2023-09-21

**Decision:**

Accept (poster)

**Comment:**

This submission received truly mixed ratings so there was a long discussion between the reviewers and authors. The rebuttal addressed some of the concerns. However, in the end, two reviewers still keep their rating as reject. The main weaknesses are practicality of the problem setting and lack of sufficient experiments. The AC took a closer look at the material and believe those weaknesses are partially addressed in the rebuttal. Then, given three positive reviews, the AC recommends accepting this paper.